# A re-inducible gap gene cascade patterns the anterior–posterior axis of insects in a threshold-free fashion

**Alena Boos[†], Jutta Distler[†], Heike Rudolf, Martin Klingler\*, Ezzat El-Sherif\***

Division of Developmental Biology, Department of Biology, Friedrich-Alexander Universität Erlangen-Nürnberg, Erlangen, Germany

**Abstract** Gap genes mediate the division of the anterior-posterior axis of insects into different fates through regulating downstream hox genes. Decades of tinkering the segmentation gene network of *Drosophila melanogaster* led to the conclusion that gap genes are regulated (at least initially) through a threshold-based mechanism, guided by both anteriorly- and posteriorly-localized morphogen gradients. In this paper, we show that the response of the gap gene network in the beetle *Tribolium castaneum* upon perturbation is consistent with a threshold-free 'Speed Regulation' mechanism, in which the speed of a genetic cascade of gap genes is regulated by a posterior morphogen gradient. We show this by re-inducing the leading gap gene (namely, *hunchback*) resulting in the re-induction of the gap gene cascade at arbitrary points in time. This demonstrates that the gap gene network is self-regulatory and is primarily under the control of a posterior regulator in *Tribolium* and possibly other short/intermediate-germ insects.
DOI: https://doi.org/10.7554/eLife.41208.001

**\*For correspondence:**
martin.klingler@fau.de (MK);
ezzat.el-sherif@fau.de (EE-S)

[†]These authors contributed
equally to this work

**Competing interests:** The
authors declare that no
competing interests exist.

**Reviewing editor:** Naama
Barkai, Weizmann Institute of
Science, Israel

## Introduction

The French Flag model is one of the earliest models of pattern formation in development (*Wolpert, 1969*), in which thresholds of a morphogen gradient (e.g. T1 and T2 in *Figure 1A'*) set the boundaries between different gene expression domains. Recent studies of morphogen-mediated patterning, however, presented several challenges to this simple picture. First, gene expression domains are usually found to be dynamic, and in many cases, are expressed sequentially in a wave-like fashion (e.g. during neural tube and limb bud patterning in vertebrates and during anterior-posterior (AP) fate specification in vertebrates and insects) (*Briscoe and Small, 2015*; *Panovska-Griffiths et al., 2013*; *Cohen et al., 2014*; *Balaskas et al., 2012*; *Dessaud et al., 2007*; *Zeller, 2004*; *Harfe et al., 2004*; *El-Sherif et al., 2014*; *El-Sherif et al., 2012a*; *Zhu et al., 2017*; *Kuhlmann and El-Sherif, 2018*). Even if activated simultaneously, gene expression domains usually undergo continuous shifts in space (e.g. gap and pair-rule domains in *Drosophila*) (*El-Sherif and Levine, 2016*; *Verd et al., 2018*; *Jaeger et al., 2004*). In such cases, it is difficult to correlate the locations of gene expression domains with specific values of morphogen concentrations. Second, morphogen exposure time was found to have a crucial effect on patterning. For example, the exposure time of Caudal (Cad) regulates the timing of gap and pair-rule genes in insects and that of Hox genes in vertebrates (*Zhu et al., 2017*; *Neijts et al., 2017*). Similarly, the concentration and exposure time of Sonic hedgehog (Shh) determines which fate a cell in the vertebrate neural tube and limb bud will take (*Dessaud et al., 2007*; *Zeller, 2004*; *Harfe et al., 2004*).

For these reasons, recent works in morphogen-mediated patterning are suggesting a more dynamic and time-based paradigm rather than threshold-based (*Briscoe and Small, 2015*; *Dessaud et al., 2007*; *Zhu et al., 2017*; *Verd et al., 2018*; *Jaeger et al., 2004*; *Verd et al., 2017*; *Clark, 2017*; *Clark and Akam, 2016*; *Clark and Peel, 2018*; *Brena and Akam, 2013*; *García-*

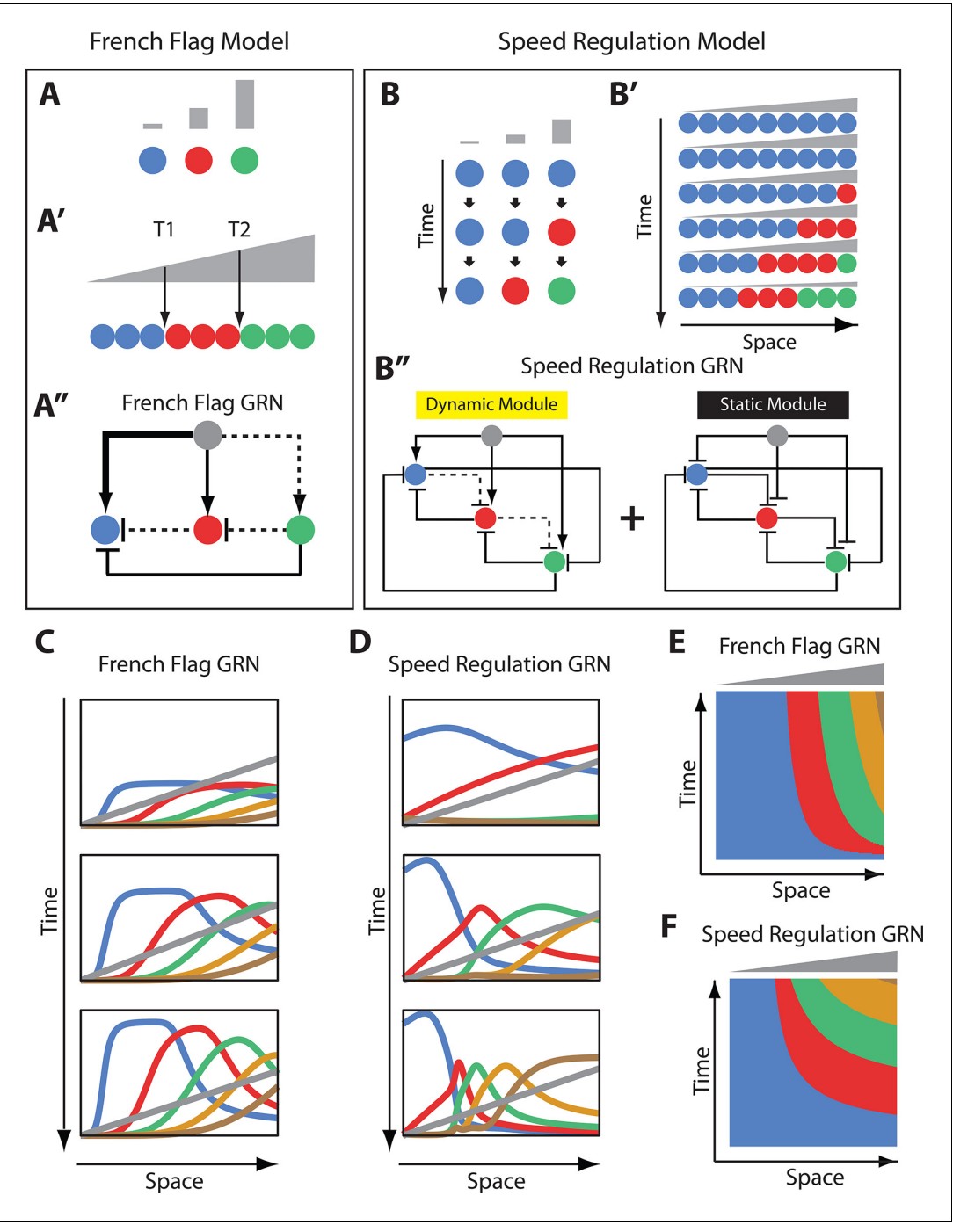

**Figure 1.** French Flag model versus Speed Regulation model. (**A–A''**). In the French Flag (FF) model, different concentrations of a morphogen gradient (grey) activate different cellular states (**A, A'**) based on a set of morphogen thresholds (here T1 and T2; **A'**). In A and A', cells are represented by circles and cellular states are shown in blue, red and green. Shown in A'' is a GRN realization of the FF model (genes representing different cellular states are shown in circles; arrowheads stand for activation, and flat bars stand for repression; the thicker the line, the stronger the activation/repression; dashed lines stand for the weakest activation/repression). (**B-B''**) In the Speed Regulation (SR) model, all cells (shown in circles in **B** and **B'**) transit through different cellular states (shown in blue, red, and green) with a speed that is proportional to the concentration of a morphogen gradient (grey). Shown in B'' is a GRN realization of the SR model (genes representing different cellular states are shown in circles; arrowheads stand for activation, and flat bars stand for repression; dashed lines stand for weak activation/repression). (**C,E**) Computer simulation of FF GRN shown as plots of expression domains along space for selected

*Figure 1 continued on next page*

*Figure 1 continued*

time points (C) and as a kymograph (E). (D,F) Computer simulation of SR GRN shown as plots of expression domains along space for selected time points (D) and as a kymograph (F).

DOI: https://doi.org/10.7554/eLife.41208.002

The following figure supplement is available for figure 1:

**Figure supplement 1.** Steady state behavior of French Flag GRN, French Flag with Timer Gene GRN, and Speed Regulation GRN.

DOI: https://doi.org/10.7554/eLife.41208.003

*Solache et al., 2010*; *Chipman and Akam, 2008*). However, the general description of the French Flag model (*Wolpert, 1969*) (*Figure 1A,A'*) is purely phenomenological (i.e. descriptive). Hence, it is unclear if a gene regulatory network (GRN) realization of the French Flag model, while still threshold-based, would reconcile the experimentally observed deviations (namely, the dynamic nature of gene expression domains and the sensitivity of patterning to morphogen exposure times). In this paper, we show that indeed important classes of GRN realization of the French Flag model 'transiently' exhibit exactly these features. In particular, at an initial transient phase, gene expression domains are dynamic and keep shifting and shrinking as long as the morphogen gradient is applied, hence its sensitivity to the exposure time of the morphogen gradient. However, gene expression domains finally stabilize at the thresholds set by the morphogen gradient, adhering to the tenets of the French Flag model. Hence, we argue that the defining feature of the French Flag model is not its transient dynamics, but its threshold-based steady state behavior.

However, a recently devised mechanism (termed 'Speed Regulation' model, *Figure 1B,B'*) (*Zhu et al., 2017*; *Kuhlmann and El-Sherif, 2018*), that has been suggested to be the basis of the AP fate specification in insects, is purely time-based and threshold-free (by 'threshold' we mean thresholds of the morphogen gradient, not other possible thresholds in the cross-regulatory interactions between patterning genes). In this model, segmentation genes of the gap class are wired into a genetic cascade, so that gap genes are activated sequentially in time. The temporal progression of gap gene expressions is translated into a spatial pattern through modulating the speed (or timing) of the gap gene cascade by a morphogen gradient of the transcription factor *caudal* (*cad*) (or of factor(s) which expression correlate with that of *cad*).

In contrast to the French Flag model, the Speed Regulation model is completely threshold-free, where gene expression domains keep shifting and shrinking as long as the morphogen gradient is applied (without ever reaching a steady state) until the gradient is retracted. In this paper, to demonstrate the threshold-free nature of gap gene regulation, we re-induce the leading gene in the gap gene cascade (namely, *hunchback*, *hb*) at arbitrary times during the AP specification phase of the beetle *Tribolium castaneum* using a transgenic line carrying a heat-shock-driven *hb* CDS. This resulted in resetting the gap gene cascade and re-establishing the gap gene expression sequence in time and space. This argues against the existence of a spatial or temporal signal that sets the locations of gap gene expression boundaries in a threshold-based fashion. Using computational modeling, we show that this self-regulatory behavior of gap gene regulation is difficult to explain using the French Flag model and, alternatively, is indicative of time-based (or 'speed regulation'-based) patterning.

The paper is organized as follows. First, we contrast the French Flag model to Speed Regulation model and their corresponding GRN realizations, highlighting their similarities and differences. We argue that the basic French Flag model, in contrast to the Speed Regulation model, is incapable of reproducing the phenomenology of insect development and evolution. Secondly, we use a modified version of the French Flag model (called 'French Flag with a Timer Gene') as a possible contender to the Speed Regulation model in explaining the AP axis fate specification in insects. We show that the major difference between the two models is that the French Flag model (and its modified version) depends on morphogen thresholds in setting the boundaries between gene expression domains, whereas the Speed Regulation model is self-regulatory and threshold-free. We suggest an experimental test to differentiate between the two mechanisms: whether the patterning scheme can be

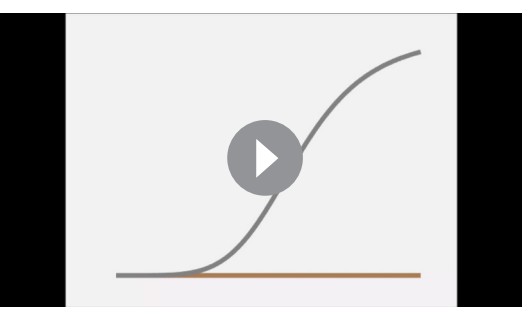 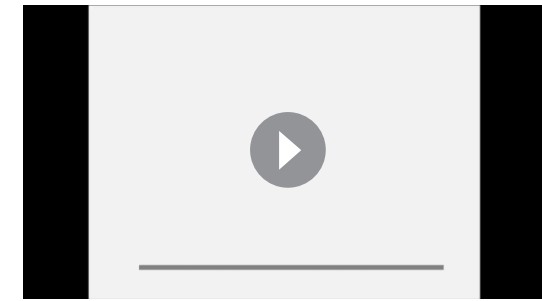

**Video 1.** Computer simulation of the French Flag GRN. A computer simulation of a 5-genes French Flag GRN (a 5-genes version of *Figure 1A''*). Genes are initially expressed in sequential waves, but their expressions eventually reach a steady state. Patterning genes are shown in blue, red, green, gold, and brown. Morphogen gradient is shown in grey. Horizontal axis is space and vertical axis is gene expression concentration.
DOI: https://doi.org/10.7554/eLife.41208.004

**Video 2.** Computer simulation of the French Flag GRN with gradient buildup dynamics. A computer simulation of a 5-genes French Flag GRN (a 5-genes version of *Figure 1A''*) with gradient buildup dynamics. Wave dynamics are more pronounced due to gradient buildup.
DOI: https://doi.org/10.7554/eLife.41208.005

reset independently from any temporal or positional signal. We then carry out this test experimentally in the beetle *Tribolium castaneum*, concluding that the Speed Regulation model is a more plausible mechanism to explain insect development and evolution.

## Results

### Comparing French Flag and Speed Regulation models in patterning non-growing tissues

A common problem in development is how to divide a group of cells into different identities, each specified by the expression of one or more genes. Two different patterning mechanisms to partition a static (i.e. non-growing) tissue along a spatial axis are shown in *Figure 1*: the French Flag (FF) model (*Figure 1A,A'*) (*Wolpert, 1969*) and the Speed Regulation (SR) model (*Figure 1B,B'*) (*Zhu et al., 2017*; *Kuhlmann and El-Sherif, 2018*). In the FF model, different ranges of a morphogen concentration (grey in *Figure 1A,A'*) activate different cellular states (specified by the expression of one or a group of genes; different states are given different colors in *Figure 1A,A'*).

In the SR model, all cells have the capacity to transit through all cellular states in the same order (blue, red, then green in *Figure 1B*). However, in contrast to the FF model, different morphogen concentrations do not directly activate specific cellular states but regulate the *speed* of state transitions in time (*Figure 1B*). Applying a gradient of the morphogen along a row of such cells induces kinematic waves of cellular states that propagate from high to low values of the gradient (*Figure 1B'*). Eventually, cells are partitioned into different domains of cellular states, and the morphogen gradient should decay and/or retract to stabilize the pattern (otherwise the expression pattern would continue to shrink and propagate towards the lower end of the gradient; last row of cells in *Figure 1B'*).

While the final results of both models are the same, their dynamics look very different. Whereas bands of cellular states in the FF model are established simultaneously without any need for a temporal component, cellular states are very dynamic in the SR model, where timing is very critical (both morphogen concentration and exposure time determine which state a cell will end up in). However, the absence of the temporal component in the French Flag model is probably unrealistic since any real life molecular implementation of the model would exhibit some transient dynamics. Hence, we aim here to use GRN realizations for both the FF and SR models and compare them on equal footing.

Using in silico evolution techniques, Francois and Siggia in ref (*François and Siggia, 2010*) performed an unbiased exploration of possible morphogen-regulated GRNs that can divide an

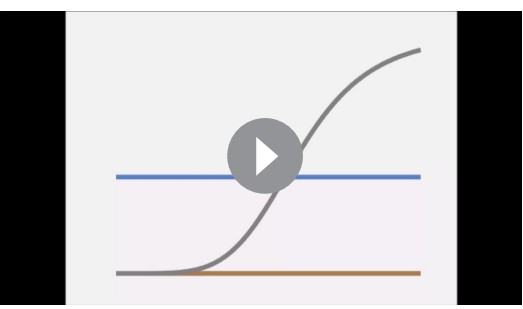

**Video 3.** Computer simulation of Speed Regulation GRN. A computer simulation of a 5-genes Speed Regulation GRN (a 5-genes version of *Figure 1B''*). Genes are expressed in sequential waves that never stabilize (except for a small region at the low end of the morphogen). Patterning genes are shown in blue, red, green, gold, and brown. Morphogen gradient is shown in grey. Horizontal axis is space and vertical axis is gene expression concentration.
DOI: https://doi.org/10.7554/eLife.41208.006

**Video 4.** Computer simulation of Speed Regulation GRN driven by a decaying morphogen gradient. A computer simulation of a 5-genes Speed Regulation GRN (a 5-genes version of *Figure 1B''*) driven by a continuously decaying morphogen gradient (grey). Genes are expressed in sequential waves that gradually stabilize due to the morphogen gradient decay. Patterning genes are shown in blue, red, green, gold, and brown. Morphogen gradient is shown in grey. Horizontal axis is space, and vertical axis is gene expression concentration.
DOI: https://doi.org/10.7554/eLife.41208.007

embryonic tissue into different fates. Although several solutions were found, they were mostly variations on the same underlying principle, which happened to be a straightforward realization of the FF model. A simple instance of this family of GRN solutions is shown in *Figure 1A''* using three genes (more genes can be added to the scheme in a straightforward manner; see Appendix 1). In the example GRN in *Figure 1A''*, the morphogen gradient (grey) activates different genes with different strengths: strongly activates the blue gene, moderately activates the red gene, and weakly activates the green gene. Cross-regulatory interaction between genes further delimit gene expression bands (See Appendix 1 for description of how the FF GRN in *Figure 1A''* works; see simulation of a 5-genes FF GRN in *Figure 1C*, *Video 1* and *Video 2*).

Now we turn to a molecular realization for the SR model recently suggested in refs (*Zhu et al., 2017*; *Kuhlmann and El-Sherif, 2018*). In this realization, two GRN modules are employed: a dynamic module and a static module (see *Figure 1B''* for a 3-genes realization and Appendix 1 for a 5-genes realization). The dynamic module is a genetic cascade that mediates the sequential activation of its constituent genes. The static module is a multi-stable network that mediates the refinement and stabilization of gene expression patterns. The morphogen gradient activates the dynamic but represses the static module. Hence, as we go from high to low values of the morphogen gradient, the dynamic module experiences excessively higher stabilizing effect from the static module, and consequently, runs slower. This is a straightforward realization of the core mechanism of the SR model (*Figure 1B,B'*) and, hence, a morphogen gradient applied to such scheme induces sequential kinematic waves that propagate from the high to the low end of the gradient as shown by the computer simulations in *Figure 1D* and *Videos 3–5*.

Comparing the spatiotemporal dynamics of the molecular realizations of the FF model and that of the SR model shows striking similarities,

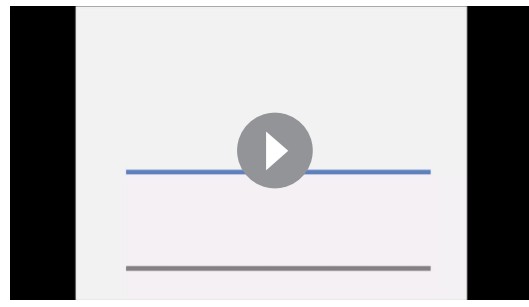

**Video 5.** Computer simulation of Speed Regulation GRN driven by a building up then decaying morphogen gradient. A computer simulation of a 5-genes Speed Regulation GRN (a 5-genes version of *Figure 1B''*) driven by a continuously building up then decaying morphogen gradient (grey).
DOI: https://doi.org/10.7554/eLife.41208.008

where gene expression domains are activated sequentially at the high end of the morphogen gradient (grey) and propagate in kinematic waves towards the low end of the gradient (compare *Figure 1C,E–1D,F* and *Videos 1–4*), especially if a morphogen buildup dynamics are introduced (compare *Video 2* and *Video 5*). Both morphogen concentration and exposure time are important factors to determine which cellular state a certain cell will have at a certain point of time (at least at the initial transient phase). A main difference, however, is that gene expression domains in the FF model realizations are only dynamic during the initial transient phase, but eventually reach a steady state where they stabilize at certain morphogen thresholds (*Figure 1—figure supplement 1A*, *Video 1*). On the other hand, gene expression domains in the SR model keep shrinking and propagating towards the low end of the morphogen and never stabilize or reach a steady state (*Figure 1—figure supplement 1C*, *Video 3*), unless the morphogen gradient decays or retracts (*Videos 4* and *5*).

## Comparing French Flag and Speed Regulation models in reproducing the phenomenology of insect development and evolution

So far, we have considered the application of the FF and SR models in patterning a static group of cells (i.e. a non-growing tissue). Here, we consider their application to the problem of insect development and evolution, where a patterning mechanism is needed to pattern both growing and non-growing tissues.

The anterior fates of insects (*Figure 2A*) form in a non-growing tissue (called the 'blastoderm'), whereas posterior fates form in a growing tissue (called the 'germband') (*Davis and Patel, 2002*; *El-Sherif et al., 2012b*). The number of fates specified in the blastoderm versus germband differ in different insects. In short-germ insects, (most of) AP fates are specified in the germband (first column in *Figure 2A*). In long-germ insects, (most of) AP fates form in the blastoderm before transiting into the germband stage (third column in *Figure 2A*). Intermediate-germ insects lie somewhere in between these two extreme cases, where several fates are specified in the blastoderm, and the rest in the germband (second column in *Figure 2A*). Throughout evolution, the specification of AP fates seems to shift easily from the germband to the blastoderm, resulting in a trend of short-germ to long-germ evolution (*Davis and Patel, 2002*; *Peel et al., 2005*; *Peel, 2004*).

In ref (*Zhu et al., 2017*), it was suggested that AP fate specification in short- and intermediate-germ insects is mediated by the SR model, where a posterior morphogen (*cad* in *Tribolium*, or some other graded factor which expression correlates with that of *cad*) regulates the sequential activation of the AP-determinant genes of the gap class. This suggestion was based on the observation that the SR model can operate in two modes (*Figure 2B*): a gradient-based, which can pattern a non-elongating tissue (as discussed in the previous section), and a wavefront-based mode in which the posterior morphogen gradient continuously retracts as the tissue elongates, in a set-up similar to the Clock-and-Wavefront model (*Pourquié, 2003*; *Palmeirim et al., 1997*; *Dubrulle et al., 2001*; *Lauschke et al., 2013*). The wavefront-based mode is best suited for patterning elongating tissues. It was also shown that the flexibility of the SR model to pattern both elongating and non-elongating tissues could offer an evolutionary mechanism where a smooth transition between short- and long-germ modes of insect development is possible (*Video 6*).

However, in ref (*François and Siggia, 2010*), Francois and Siggia suggested a simple modification that enables also the FF model to exhibit such flexibility in patterning both elongating and non-elongating tissues. In this scheme, the posterior morphogen (fourth column in *Figure 2A*, and grey in *Figure 2C*) activates a gene (termed a 'Timer Gene' (*François and Siggia, 2010*); fifth column in *Figure 2A*, and black in *Figure 2C*). The Timer Gene is assumed to have negligible decay rate, so that it continuously builds up in a non-growing tissue (left column in *Figure 2C*). In a growing tissue, the expression of the Timer Gene (black in *Figure 2C*, right column) builds up in the presence of the retracting posterior gradient (grey in *Figure 2C*, right column), while it stabilizes upon its retraction. Hence, a long-range gradient of the Timer Gene forms along the whole axis of the full-grown tissue (last row in the fifth column of *Figure 2A* and the right column of *Figure 2C*). Thresholds of different concentrations of the Timer Gene then set the boundaries between different fates, in a similar fashion to the FF model (thresholds are shown in arrows in *Figure 2C*). We call this scheme 'French Flag model with a Timer Gene' (FFTG). We notice that both the SR model and the FFTG model can operate in gradient-based and wavefront-based modes (and, hence, can pattern both non-elongating

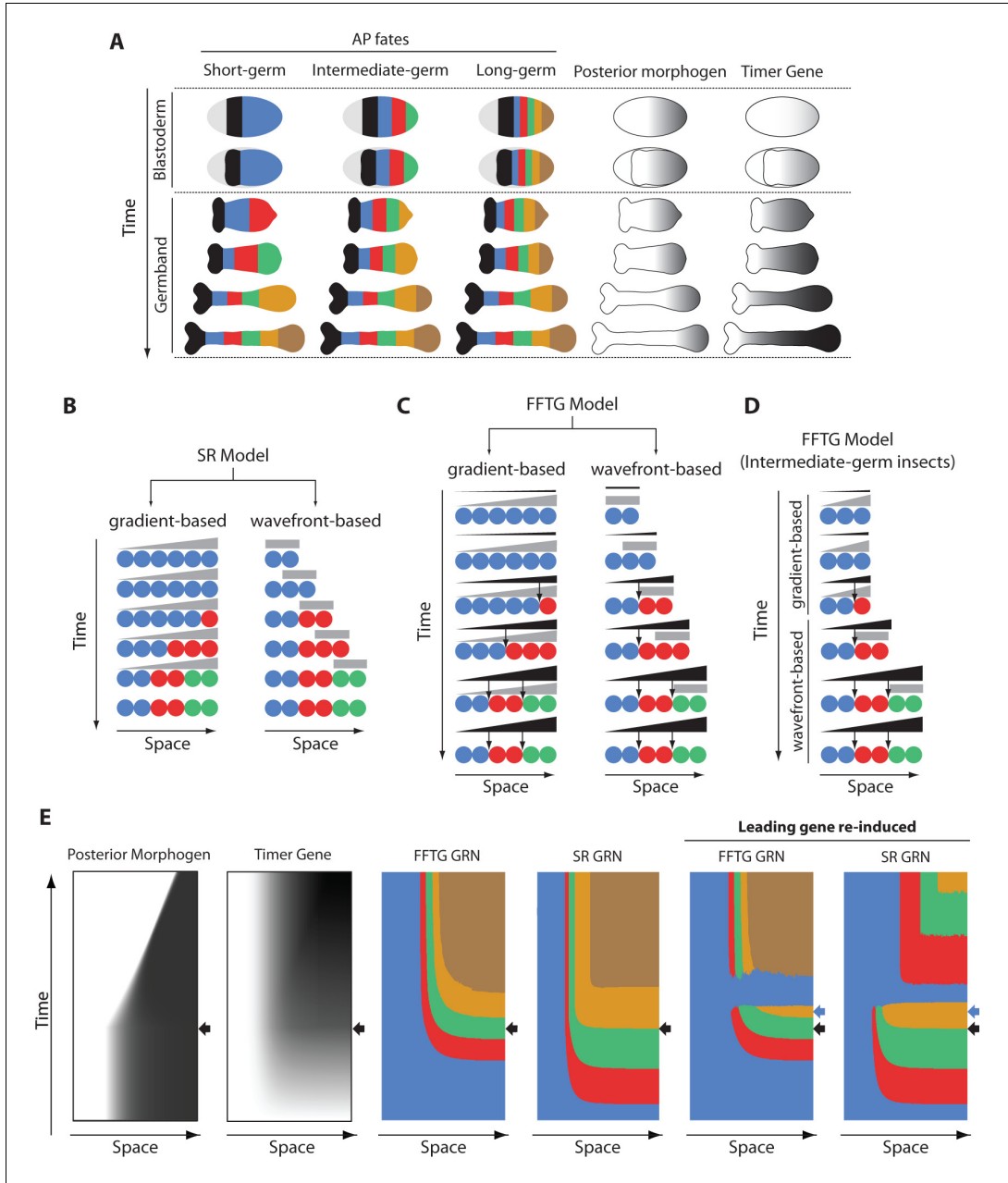

**Figure 2.** Application of FF and SR model to the problem of insect development and evolution. (**A**) Anterior-posterior (AP) fate specification in insects of different germ types: short-, intermediate- and long-germ. Different fates are shown in different colors. In short-germ embryogenesis, most of fates are specified in an elongating embryonic structure called 'germband'. In long-germ embryongenesis, most of fates are specified in a non-elongating embryonic structure called 'blastoderm'. In intermediate-germ embryogenesis, anterior fates are specified in the blastoderm, while posterior fates are specified in the germband. A posteriorly-localized morphogen is shown in grey. Timer Gene, as required by French Flag with Timer Gene (FFTG) model is shown in black with different shades (the darker the higher the concentration). (**B**) The Speed Regulation (SR) model can operate in two modes: gradient-based and wavefront-based. In the gradient-based mode, a static gradient of the speed regulator (grey) is applied to a static field of cells (shown in circles). In the wavefront-based mode, a boundary of the speed regulator retracts along an elongating field of cells. (**C**) The FFTG model can operate in a gradient-based or a wavefront-based mode as well. The Timer Gene (black) is activated by a posterior morphogen (grey) and suffers little or no decay so that it unfolds into a long-range gradient along the entire AP axis in either the gradient-based or the wavefront-based mode. Different concentrations of the Timer Genes (arrows) set the boundaries between different fates. (**D**) FFTG model can pattern the AP axis of intermediate-germ insects: acting

*Figure 2 continued on next page*

*Figure 2 continued*

in the gradient-based mode to pattern anterior fates (during the blastoderm stage) and acting in the wavefront-based mode to pattern posterior fates (during the germband stage). (**E**) Kymographs of FFTG and SR GRNs when applied to AP patterning of intermediate-germ insects (first four panels; black arrow marks blastoderm-to-germband transition). Panels 5 and 6 show the response of the FFTG and SR GRNs to the re-activation of the leading gene (blue; the time of blue gene re-induction is marked by a blue arrow). In FFTG GRN, normal expression pattern is restored after a brief dominance of the blue gene. In SR GRN, already formed expression is down-regulated and the genetic cascade is reset.

DOI: https://doi.org/10.7554/eLife.41208.009

---

and elongating tissues) and exhibit similar dynamics and final pattern (compare *Figure 2B and C*; see *Figure 1—figure supplement 1*).

A transition from a gradient-based to wavefront-based patterning would still result in a long-range gradient of the Timer Gene along the tissue axis (*Figure 2D*). Hence, similar to the SR model, the FFTG model can mediate AP fate specification in insects of different germ types: operating in the wavefront-based mode for short-germ insects, operating in the gradient-based mode for long-germ insects, and operating in the gradient-based then switching to wavefront-based mode for intermediate-germ insects (*Video 7*; compare to *Video 6*).

Since gene expression dynamics of both the SR and FFTG models are very similar, we sought an experimental test to differentiate between the two models. In our demonstration of the experimental test, we will use the GRN realization of both SR and the FFTG models (5-genes version of

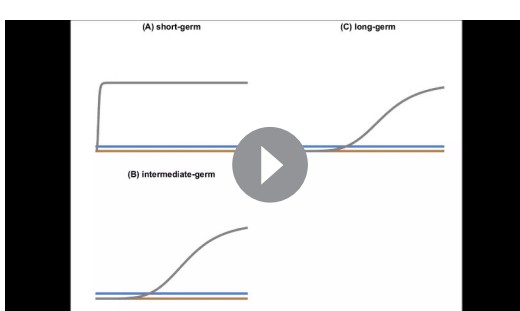

**Video 6.** Speed Regulation GRN patterning the Anterior-Posterior axis of short-germ, intermediate-germ and long-germ insects. A computer simulation of fate specification in insects with different germ types using a 5-genes Speed Regulation GRN. (**A**) In short-germ insects, the posterior morphogen (grey) continuously retracts towards posterior with axis elongation. (**B**) In intermediate-germ insects, the posterior morphogen is initially expressed in a static gradient that eventually retracts towards posterior with axis elongation. (**C**) In long-germ insects, the posterior morphogen is expressed in a static gradient throughout the patterning process. Patterning genes are shown in blue, red, green, gold, and brown. Posterior morphogen gradient is shown in grey. Horizontal axis represents the Anterior-Posterior axis. Posterior to the right. Vertical axis is gene expression concentration.

DOI: https://doi.org/10.7554/eLife.41208.010

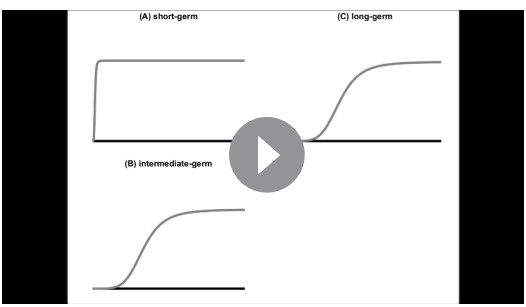

**Video 7.** French Flag with a Timer Gene GRN patterning the Anterior-Posterior axis of short-germ, intermediate-germ and long-germ insects. A computer simulation of fate specification in insects with different germ types using a 5-genes French Flag with a Timer Gene GRN. (**A**) In short-germ insects, the posterior morphogen (grey) continuously retracts towards posterior with axis elongation. (**B**) In intermediate-germ insects, the posterior morphogen is initially expressed in a static gradient that eventually retracts towards posterior with axis elongation. (**C**) In long-germ insects, the posterior morphogen is expressed in a static gradient throughout the patterning process. Patterning genes are shown in blue, red, green, gold, and brown. Posterior morphogen gradient is shown in grey. Timer Gene is shown in black. Horizontal axis represents the Anterior-Posterior axis. Posterior to the right. Vertical axis is gene expression concentration.

DOI: https://doi.org/10.7554/eLife.41208.011

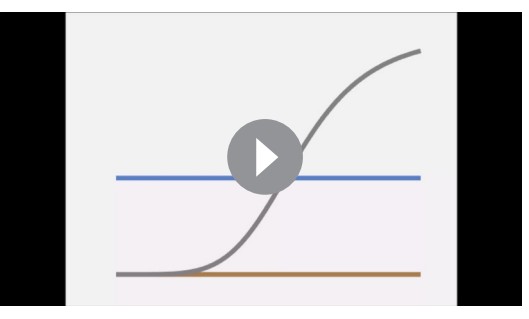 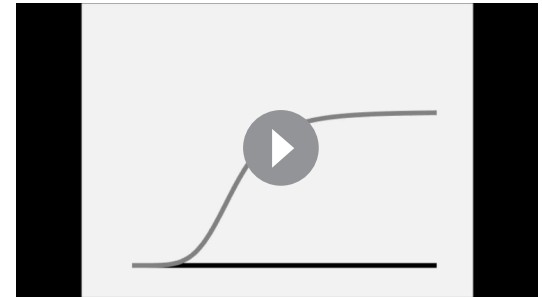

**Video 8.** Re-inducing the leading gene in the Speed Regulation GRN during intermediate-germ patterning. Re-inducing the leading gene (blue) in the Speed Regulation GRN during a simulation of intermediate-germ patterning results in dual response: already established genes in the anterior are down-regulated, while the sequential activation of genes is reset within the expression domain of the posterior morphogen (grey). Patterning genes are shown in blue, red, green, gold, and brown. Posterior morphogen gradient is shown in grey. Horizontal axis represents the Anterior-Posterior axis. Posterior to the right. Vertical axis is gene expression concentration.

DOI: https://doi.org/10.7554/eLife.41208.012

**Video 9.** Re-inducing the leading gene in the French Flag with a Timer Gene GRN during intermediate-germ patterning. Re-inducing the leading gene (blue) in the French Flag with a Timer Gene GRN during a simulation of intermediate-germ patterning results in a transient dominance of the leading gene, but eventual formation of the normal gene expression pattern. Patterning genes are shown in blue, red, green, gold, and brown. Posterior morphogen gradient is shown in grey. Horizontal axis represents the Anterior-Posterior axis. Posterior to the right. Vertical axis is gene expression concentration.

DOI: https://doi.org/10.7554/eLife.41208.013

*Figure 1B''* and *5*-genes version of *1A''* after adding a Timer Gene, respectively; see Appendix 1). We apply both models to the case of AP fate specification of an intermediate-germ insect.

Here, we note that the sequential activation of genes in the SR model is mediated by the interaction between the fate-specifying genes themselves, whereas the Timer Gene is the mediator of sequential gene activation in the FFTG model. Hence, force-resetting the fate-specification gene sequence will reset the SR model, whereas resetting the expression pattern for the FFTG model would require resetting the Timer Gene instead. This is evident in our simulations of both models in *Figure 2E* and *Videos 8* and *9*. After the expression of the first three AP fate-specifying genes (blue, red, and green in *Figure 2E*), the blue gene was briefly re-induced. In the FFTG model, the blue gene becomes briefly dominant, but the already formed pattern and the newly forming pattern are largely unchanged (compare 5th to 3rd column of *Figure 2E*). This is a natural consequence of the fact that the Timer Gene (*Figure 2E*, second column) is the main driver of the patterning process, which is unaffected by the blue gene re-induction. On the other hand, re-inducing the blue gene had two consequences in the case of our GRN realization of the SR model (compare 4th and 6th columns of *Figure 2E*): (*i*) the already formed pattern outside of the expression domain of the posterior morphogen is deleted and dominated by the continued expression of the blue gene, and (*ii*) the temporal gene sequence is re-established within the expression of the posterior morphogen, resulting in the re-establishment of the patterning process. This dual effect results from the dual regulation mode of our realization of the SR model: a dynamic genetic module is active within the posterior morphogen expression domain, whereas a static module is active outside. In our GRN realization, re-inducing the blue gene resets the dynamic module (which is basically a genetic cascade), while it down-regulates all the genes of the static module except the re-induced blue gene (since it is a multi-stable mutually exclusive GRN).

## A dual response upon re-inducing *hunchback* in the *Tribolium* embryo

During AP patterning of the *Tribolium* embryo, gap genes (namely, *hunchback* (*hb*), *Krüppel* (*Kr*), *milles-pattes* (*mlpt*), and *giant* (*gt*); *Figure 3*) (*Bucher and Klingler, 2004*; *Wolff et al., 1998*; *Savard et al., 2006*; *Cerny et al., 2005*; *Marques-Souza, 2007*) are expressed in sequential waves of gene expressions that propagate from posterior to anterior in the presence of a gradient of the master regulator *caudal* (*cad*) (based on a correlational evidence, however) (*Zhu et al., 2017*).

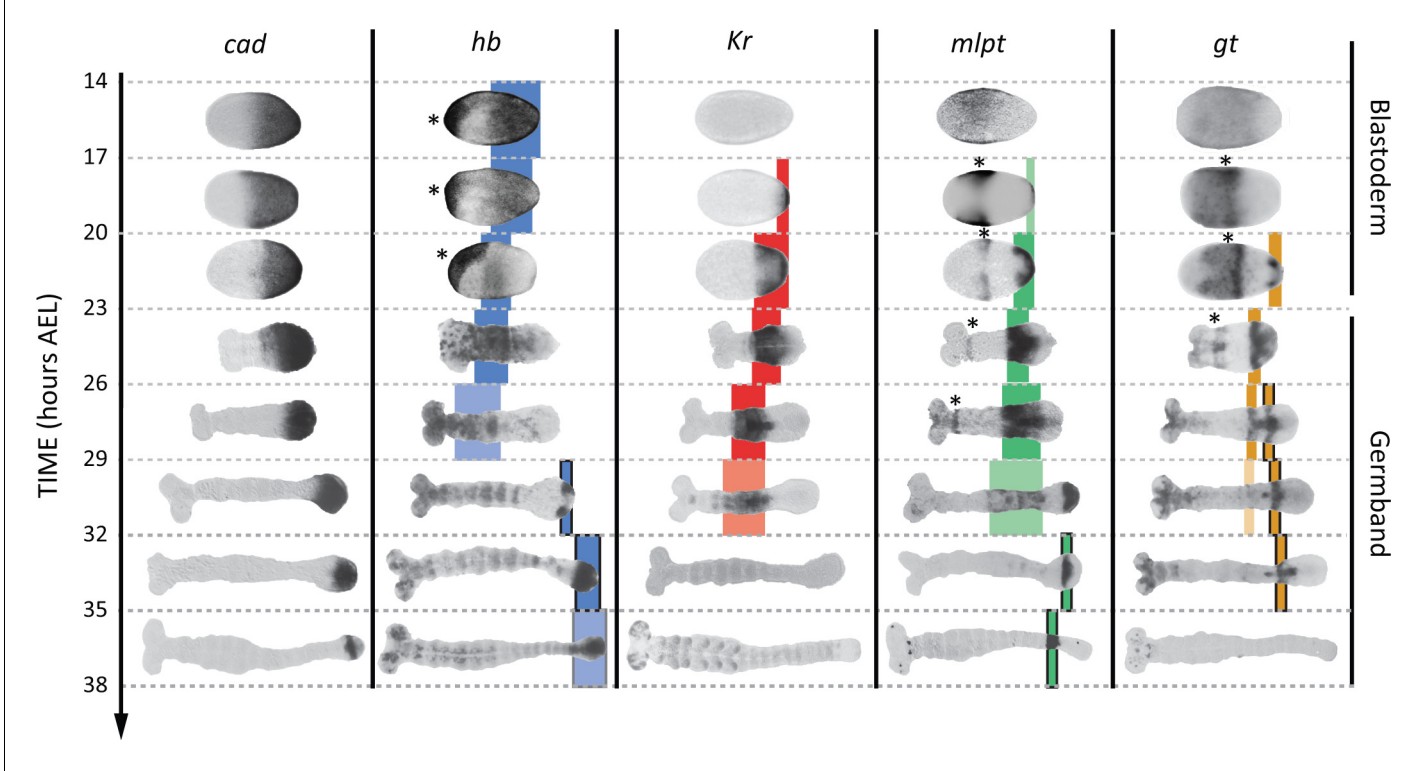

**Figure 3.** Expression dynamics of *caudal* and gap genes during AP axis specification in *Tribolium*. *caudal* is expressed in a static (i.e. non-retracting) posterior-to-anterior gradient during the blastoderm stage, while is expressed in a retracting posterior-to-anterior gradient during the germband stage. Gap genes are expressed in sequential waves of gene expressions that propagate from posterior to anterior. Expression of different gap genes are tracked with differed colors: blue for *hb*, red for *Kr*, green for *mlpt*, and gold for *gt*. The second trunk domains of *hb*, *mlpt* and *gt* are outlined in black. Weak expressions are shown in faint colors. Non-trunk and extraembryonic expressions of gap genes (not considered in our analysis) are marked with asterisks. Posterior to the right in all embryos shown.

DOI: https://doi.org/10.7554/eLife.41208.014

Hereafter, we will call the region of the embryo where *cad* is expressed: the 'Active-Zone'. Upon retraction of the *cad* gradient, gap gene expressions stabilize into static domains before they gradually fade. Some of the gap genes have two trunk expression domains, namely: *hb*, *mlpt* and *gt* (shown in blue, green and gold, respectively in *Figure 3*; late trunk domains are outlined in black).

To determine whether the FF or the SR model is involved in regulating gap genes in *Tribolium*, we sought to re-induce the first gene in the gap gene sequence, namely *hb*, at arbitrary times during AP patterning in the *Tribolium* embryo. To this end, we constructed a transgenic line carrying a *hb* CDS under the control of a heat-shock promoter (hs-hb line; see Materials and methods and *Figure 4—figure supplement 1*). Briefly heat-shocking hs-hb embryos at 26–29 hr After Egg Lay (AEL) indeed resulted in a ubiquitous expression of *hb* that lasted for 6 hr post heat-shock (*Figure 4—figure supplement 2*; for a basic description of the cuticular and morphological phenotypes of heat-shocked hs-hb embryos, see Appendix 1). As a control, we also heat-shocked WT embryos at 26–29 hr AEL, and noticed a normal progression of gap gene expression, albeit with an initial delay of around 9 hr, compared to non-heat-shocked WT embryos (compare *Figure 4A* and *Figure 4—figure supplement 2* to *Figure 3*; see also *Figure 4C*).

Next, we examined and contrasted the expressions of the other gap genes (*Kr*, *mlpt*, and *gt*) in heat-shocked hs-hb and heat-shocked WT embryos. We observed that cells in hs-hb embryos had two distinct responses depending on whether they are within or anterior to the active-zone. Gene expression domains anterior to the active-zone are pre-maturely repressed, compared to heat-shocked WT. Within the active-zone, the gap gene domains sequence is re-induced, and eventually propagates towards anterior (*Figure 4A*). Specifically, in heat-shocked hs-hb embryos, at around 35–38 hr AEL, *Kr* expression anterior to the active-zone was down-regulated (*Figure 4A*). At around

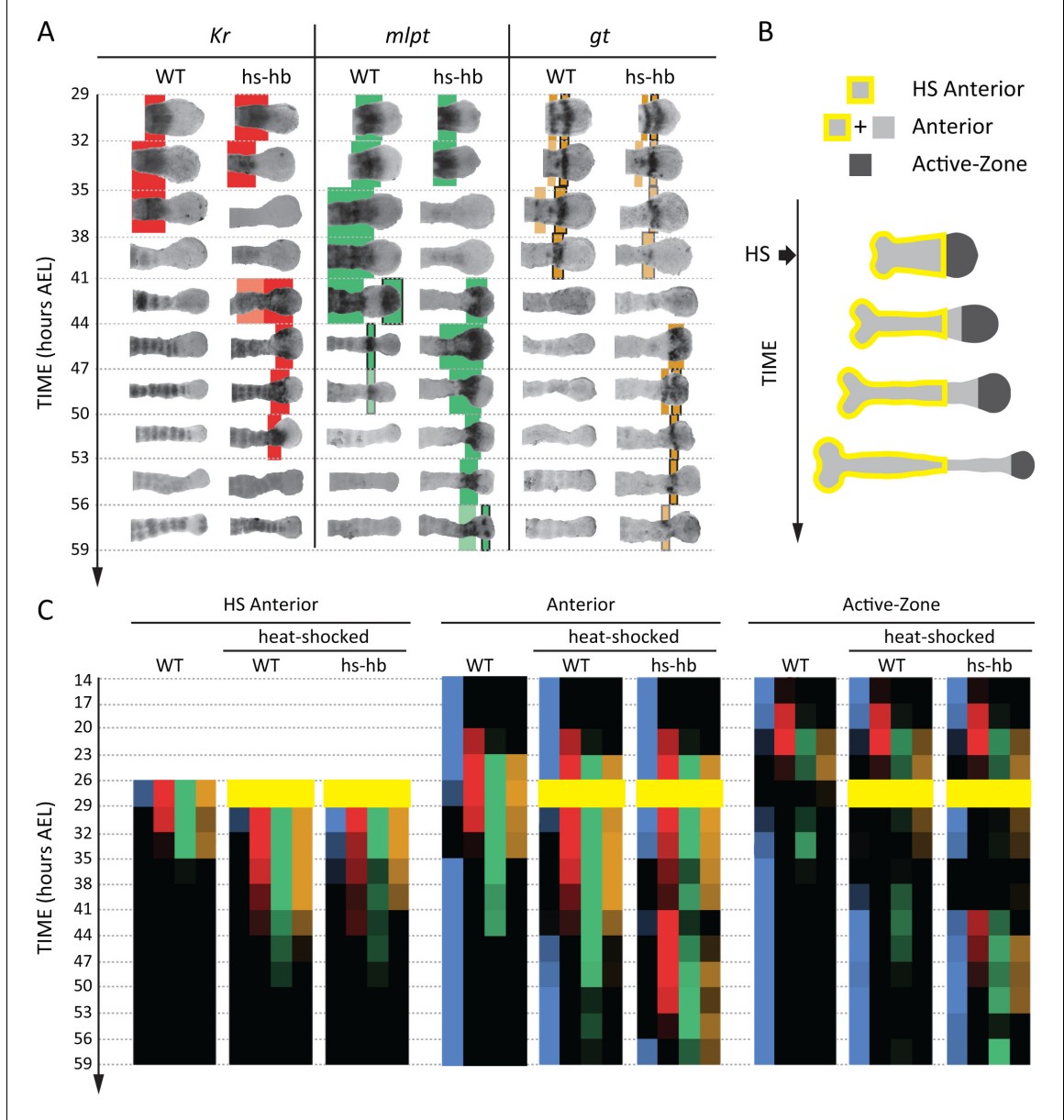

**Figure 4.** Dual response of *Tribolium* embryos upon re-inducing *hb* expression at 26–29 hr AEL. (**A**) Expression dynamics of gap genes *Kr*, *mlpt*, and *gt* upon heat-shocking both WT and hs-hb embryos at 26–29 hr AEL. Shown are posterior halves of embryos (see *Figure 4—figure supplement 2* for whole embryos). Expression of different gap genes are tracked with differed colors: red for *Kr*, green for *mlpt*, and gold for *gt*. The second trunk domains of *mlpt* and *gt* are outlined in black. Weak expressions are shown in faint colors. Posterior to the right in all embryos shown. (**B**) Dividing heat-shocked *Tribolium* embryos into three domains: Active-Zone (*caudal*-expressing zone at the posterior end of the embryo), Anterior (anterior to the Active-Zone), and HS Anterior (anterior to the Active-Zone at the time of applying heat-shock). See Materials and methods for a description of the used morphological landmarks to differentiate between these three regions. (**C**) Quantification (see Materials and methods) of gap gene expressions in WT, heat-shocked WT, and heat-shocked hs-hb embryos (heat-shocks applied at 26–29 hr AEL). Quantifications are carried out separately for HS Anterior, Anterior, and Active-Zone. While heat-shock application resulted in only a temporal delay in gap gene expression in WT, it resulted in dual response for hs-hb embryos: already established gap gene domains are pre-maturely down-regulated in HS Anterior, while the gap gene sequence is re-activated in Active-Zone. Re-induced gap gene sequence eventually propagates into Anterior. Time-windows where heat-shock is applied are shown in yellow.

DOI: https://doi.org/10.7554/eLife.41208.015

The following source data and figure supplements are available for figure 4:

**Source data 1.** Raw numerical data of quantitative analyses for embryos heat-shocked at 26–29 hr AEL.
DOI: https://doi.org/10.7554/eLife.41208.019

**Figure supplement 1.** Map of the overexpression vector pBac[hsp68-dsRed-hsp68-hb].

*Figure 4 continued on next page*

*Figure 4 continued*

DOI: https://doi.org/10.7554/eLife.41208.016

**Figure supplement 2.** Dual response of *Tribolium* embryos upon re-inducing *hb* expression at 26–29 hr AEL.

DOI: https://doi.org/10.7554/eLife.41208.017

**Figure supplement 3.** Quantification of the response of *Tribolium* embryos upon re-inducing *hb* expression at 26–29 hr AEL.

DOI: https://doi.org/10.7554/eLife.41208.018

41 hr AEL, *Kr* expression was re-initiated in the active-zone of heat-shocked hs-hb embryos, an effect that is not noticed in heat-shocked WT embryos (*Figure 4A*). The re-initiated *Kr* expression then propagated towards anterior. A similar effect is observed for the gap gene *mlpt*. At 35 hr AEL, the already established *mlpt* expression at the anterior of hs-hb embryos was down-regulated. By 41 hr AEL, *mlpt* expression was re-established in the active-zone and propagated towards anterior. The second domain of the re-established *mlpt* expression appeared at 56 hr AEL. Similarly, the already formed two domains of *gt* expression were repressed in the anterior and new two domains of expressions were re-established in the posterior of hs-hb embryos that eventually propagated towards anterior (*Figure 4A*).

We then characterized the response of the *Tribolium* embryo upon *hb* re-induction more quantitatively within and anterior to the active-zone. Since cells within the active-zone progressively move out of this region during axis elongation, we performed our analysis for three regions (*Figure 4B*; see Materials and methods): (*i*) the active-zone, (*ii*) anterior to the active-zone at the time of heat-shock application (hereafter called 'HS Anterior'), and (*iii*) anterior to the active-zone at the time of analysis (hereafter called the 'Anterior', which includes HS Anterior and the cells that have moved out of the active-zone since the time of heat-shock application (see Materials and methods for the morphological markers used to differentiate between these three regions). By examining the distribution of gene activities within these three regions (Active-Zone, HS Anterior, and Anterior) over time, we confirmed the dual response of *Tribolium* embryos upon re-inducing *hb* (*Figure 4C*; See *Figure 4—figure supplement 3* for error bars; see *Figure 4—source data 1* for source data and sample sizes). In HS Anterior cells, gap genes are pre-maturely down-regulated; while within the Active-Zone, gap gene sequence is re-induced and eventually propagates towards the Anterior region (as shown in HS Anterior, Active-Zone, and Anterior, respectively in *Figure 4C*).

It is worth noting here that, in WT, starting from 14 hr AEL, *hb*, *Kr*, *mlpt* and *gt* expression domains arise sequentially in the active-zone (*Figure 3*). By 26 hr AEL, the majority of gap gene expression domains already propagated out of the active-zone towards anterior, and the active-zone becomes virtually void of any gap gene expression. Around 29 hr AEL, the second abdominal domains of *mlpt* and *hb* arise in the active-zone (*Figure 3*). In *Figure 4*, we performed our heat-shock experiments within the 26–29 hr AEL time window when the active-zone is void of gap gene expression. The fact that the entire gap gene sequence is re-induced upon re-inducing *hb* in the active-zone at a point in time and space where gap genes are not expressed in WT indicates that gap genes are wired into an 'aperiodic' clock that can be reset at any point in time and strongly argues against a threshold-based French Flag model underlying gap gene regulation in *Tribolium* and is consistent with a mechanism based on the Speed Regulation model, as discussed in our earlier theoretical analysis. Furthermore, the dual response of the *Tribolium* embryo to the re-induction of *hb* provides an indirect support for the two-modules GRN realization of the SR model (*Figure 1B''*). The difference in response between the active-zone and anterior cells can be explained by a difference in the genetic wiring of gap genes in these two regions, encoded by two different GRNs (e.g. dynamic and static modules).

So far, we considered the effects of re-inducing *hb* at 26–29 hr AEL, a time window where the active-zone is void of gap gene expressions. This helped avoiding a possible interference between gap gene expressions already present in the active-zone and the re-induced expressions. To investigate the outcome of re-inducing the gap gene sequence while the original gene expression sequence is unfolding, we preformed our heat-shock experiments also at time windows 23–26 hr AEL (*Figure 5A* and *Figure 5—figure supplement 1*; See *Figure 5—figure supplement 2* for error bars; see *Figure 5—source data 1* for source data and sample sizes) and 20–23 hr AEL (*Figure 5B* and *Figure 5—figure supplement 3*; See *Figure 5—figure supplement 4* for error bars; see *Figure 5—source data 2* for source data and sample sizes). For both time windows, the expression

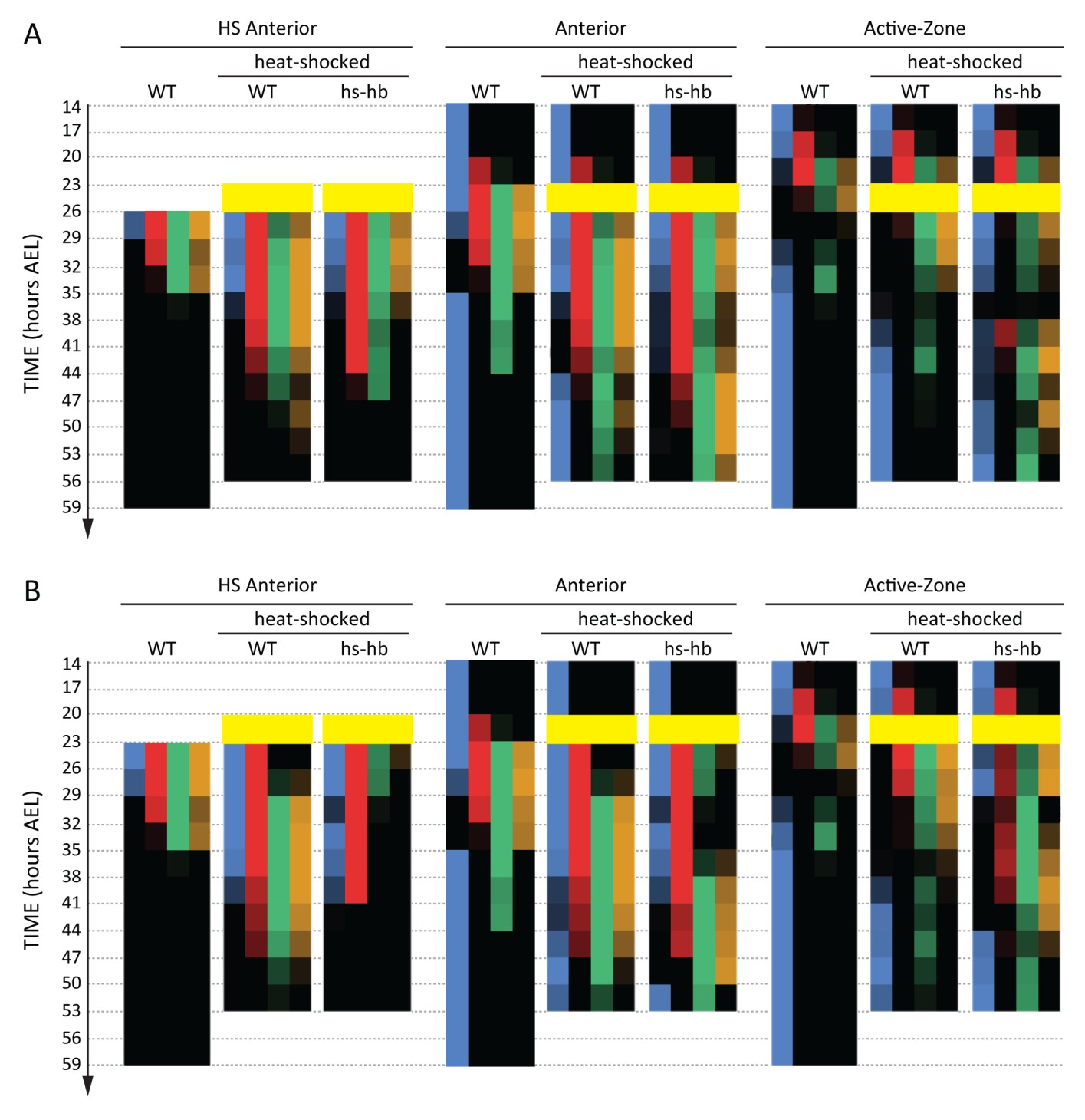

**Figure 5.** Dual response of *Tribolium* embryos upon re-inducing *hb* expression at 23–26 and 20–23 hr AEL. Quantification (see Materials and methods) of gap gene expressions in WT, heat-shocked WT, and heat-shocked hs-hb embryos. Heat-shocks are applied at 23–26 hr AEL (**A**) and 20–23 hr AEL (**B**). Quantifications are carried out separately for HS Anterior, Anterior, and Active-Zone. While heat-shock application resulted in only a temporal delay in gap gene expression in WT, it resulted in dual response for hs-hb embryos: already established gap gene domains are pre-maturely down-regulated in HS Anterior, while the gap gene sequence is re-activated in Active-Zone. Re-induced gap gene sequence eventually propagates into Anterior. Time-windows where heat-shock is applied are shown in yellow.

DOI: https://doi.org/10.7554/eLife.41208.020

The following source data and figure supplements are available for figure 5:

*Figure 5 continued on next page*

*Figure 5 continued*

**Source data 1.** Raw numerical data of quantitative analyses for embryos heat-shocked at 23–26 hr AEL.
DOI: https://doi.org/10.7554/eLife.41208.025
**Source data 2.** Raw numerical data of quantitative analyses for embryos heat-shocked at 20–23 hr AEL.
DOI: https://doi.org/10.7554/eLife.41208.026
**Figure supplement 1.** Dual response of *Tribolium* embryos upon re-inducing *hb* expression at 23–26 hr AEL.
DOI: https://doi.org/10.7554/eLife.41208.021
**Figure supplement 2.** Quantification of the response of *Tribolium* embryos upon re-inducing *hb* expression at 23–26 hr AEL.
DOI: https://doi.org/10.7554/eLife.41208.022
**Figure supplement 3.** Dual response of *Tribolium* embryos upon re-inducing *hb* expression at 20–23 hr AEL.
DOI: https://doi.org/10.7554/eLife.41208.023
**Figure supplement 4.** Quantification of the response of *Tribolium* embryos upon re-inducing *hb* expression at 20–23 hr AEL.
DOI: https://doi.org/10.7554/eLife.41208.024

within the active-zone suffered a transient down-regulation before the gap gene sequence is re-induced. This indicates that the gap gene clock is based on a genetic cascade with mutual-repressive links, like the one we used in our theoretical analysis (dynamic module in *Figure 1B''*).

## The re-induction of gap gene sequence upon re-inducing *hb* is specific to the active-zone

In our earlier theoretical analysis (*Zhu et al., 2017*), we considered a realization of the SR model that relies on the gradual switching between two genetic modules. An alternative realization would be to jointly regulate the activation and degradation rates of gap genes by a posterior morphogen gradient (computational modeling in Appendix 1; *Video 10*). A major prediction of the module switching model in the case of gap gene regulation in *Tribolium* is that the genetic wiring of gap genes in the presence of the morphogen *cad* (i.e. within the active-zone) is different from their wiring in the absence of *cad* (i.e. anterior to the active-zone). As discussed above, the difference in response to the re-induction of *hb* between the active-zone and the anterior supports the module switching model (and disfavors a degradation rate modulation model; *Video 11*). To further test this, we sought to examine if the transient down-regulation and the subsequent re-activation of gap genes is specific to the active-zone, i.e. the region of the embryo expressing *cad*. To this end, we utilized the *axin* (*axn*) RNAi phenotype in *Tribolium*, where the *cad* gradient extends to cover most of the embryo, transforming the embryo into an enlarged active-zone (albeit still expressed in a gradient; *Figure 6—figure supplement 1*) (*Zhu et al., 2017*; *Fu et al., 2012*). We performed our heat-shock experiments at 20–23 hr AEL time window for WT embryos, hs-hb embryos, embryos laid by WT mothers injected with *axn* dsRNA (*axn* RNAi embryos), and embryos laid by hs-hb mothers injected with *axn* dsRNA (hs-hb; *axn* RNAi embryos). We then analyzed the expression of *mlpt* at consecutive 3 hr time windows starting from 32 hr AEL. As shown earlier, in hs-hb embryos, *mlpt* initially suffered a transient down-regulation. Shortly after, *mlpt* expression is re-established within the active-zone. In *axn* RNAi embryos, *mlpt* expression proceeded as in WT but propagated across the whole embryo and never stabilized, consistent with the fact that the whole embryo transformed into an active-zone (*Zhu et al., 2017*). In hs-hb; *axn* RNAi embryos, after a transient down-regulation of *mlpt* expression, *mlpt* expression emerged at the posterior then expanded to cover the whole embryo, an effect only observed in the active-zone in WT embryos. This effect is recapitulated in a computer simulation of the *axn* phenotype (*Video 12*; Appendix 1). This supports the hypothesis that the re-activation of gap gene sequence in *Tribolium* upon re-inducing *hb* is specific to the region of the embryo where the posterior morphogen *cad* is expressed.

## Discussion

In this paper, we argued that an important class of GRN realizations of the FF model exhibits dynamic gene expression patterns and are sensitive to morphogen exposure time, at least during an initial transient phase. Nonetheless, such realization is faithful to the essence of the FF model where morphogen concentration thresholds set the boundaries between different gene expression domains. These thresholds are inscribed either explicitly by tuning the binding affinities of the

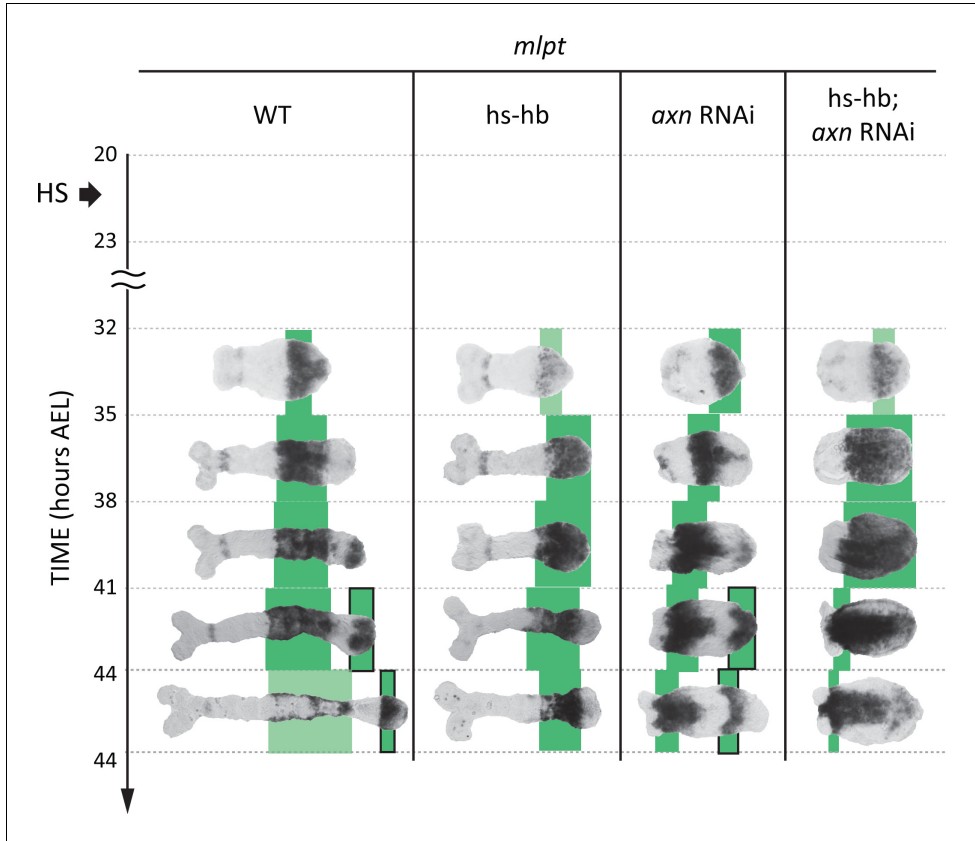

**Figure 6.** Gap gene sequence re-activation upon re-inducing *hb* is specific to the active-zone. *mlpt* expression is re-activated only in the active-zone (posterior *cad*-expressing region) upon heat-shocking hs-hb embryos (compare WT and hs-hb embryos). Knocking-down *axn* completely posteriorized *Tribolium* embryos (*axn* RNAi) such that nearly the entire embryo becomes a big active-zone (as evident from *cad* expression; see **Figure 6—figure supplement 1**), where *mlpt* expression is very dynamic and propagates across the entire embryo. Upon heat-shocking hs-hb embryos whose mothers had been injected with *axn* dsRNA (hs-hb; *axn* RNAi), *mlpt* expression is first activated at the posterior then propagates to cover the whole embryo, supporting the hypothesis that gap gene re-activation in specific to the *cad*-expressing domain (the active-zone). *mlpt* expresssion is tracked in green. The second trunk domain of *mlpt* is outlined in black. Weak expressions are shown in faint colors. Posterior to the right in all embryos shown.

DOI: https://doi.org/10.7554/eLife.41208.029

The following figure supplement is available for figure 6:

**Figure supplement 1.** Knocking-down *axn* by RNAi posteriorizes *Tribolium* embryos.

DOI: https://doi.org/10.7554/eLife.41208.030

morphogen gradient to the *cis*-regulatory elements of downstream genes (like the FF GRN we presented in this study), implicitly by tuning cross-regulatory strengths between genes (like the AC-DC motif that was found to be involved in patterning the ventral neural tube of vertebrates (*Panovska-Griffiths et al., 2013*; *Balaskas et al., 2012*; *Perez-Carrasco et al., 2018*)) or using a combination of both strategies. In contrast, the SR model is essentially threshold-free and drives ever-changing gene expression patterns as long as the morphogen gradient is applied. We also considered a modified version of the FF model (the FFTG model) that, while exhibiting similar spatiotemporal dynamics as SR model, is still threshold-based.

In fact, the SR and the FFTG models are more intimately related than it first appears. Both models can be thought of as composed of an 'aperiodic' clock whose speed is regulated by the posterior morphogen (grey in **Figure 2B and C**). The main difference between the two mechanisms is the nature of the clock. In the SR model, the clock is mediated by the regulatory interactions between the fate-specifying genes themselves (e.g. the dynamic module in **Figure 1B''**). Each tick of the clock

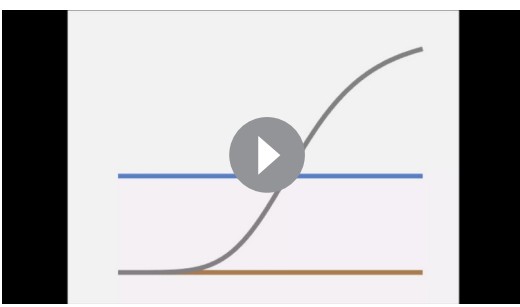

**Video 10.** Simulation of GRN realizing the Speed Regulation model by jointly modulating gene activity and gene product decay rates. A computer simulation of the Speed Regulation model realized by jointly modulating gene activity and gene products decay rates (described in Appendix 1) applied to the problem of patterning the anterior-posterior axis of an intermediate-germ insect. Patterning genes are shown in blue, red, green, gold, and brown. Posterior morphogen gradient is shown in grey. Horizontal axis represents the Anterior-Posterior axis. Posterior to the right. Vertical axis is gene expression concentration.
DOI: https://doi.org/10.7554/eLife.41208.027

is specified by a different (combination of) fate specifying gene(s). In the suggested molecular realization of the SR model, the speed of the

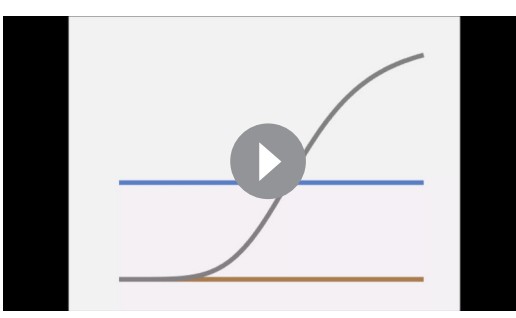

**Video 11.** Re-inducing the leading gene in the Speed Regulation GRN realization by jointly modulating gene activity and gene products decay rates during intermediate-germ patterning. Re-inducing the leading gene (blue) in the Speed Regulation GRN realization by jointly modulating gene activity and gene products decay rates during a simulation of intermediate-germ patterning results resetting the sequential activation of patterning genes within the expression of the posterior morphogen (grey) but leaves the already established gene expression in the anterior intact. Patterning genes are shown in blue, red, green, gold, and brown. Posterior morphogen gradient is shown in grey. Horizontal axis represents the Anterior-Posterior axis. Posterior to the right. Vertical axis is gene expression concentration.
DOI: https://doi.org/10.7554/eLife.41208.028

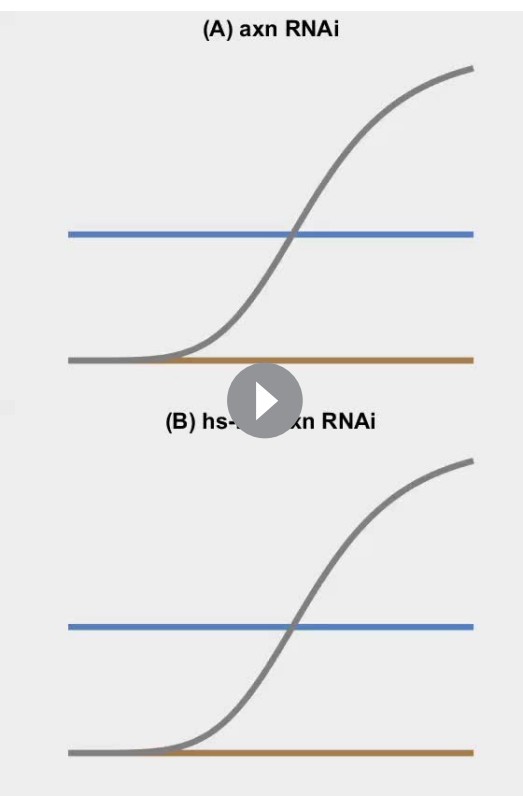

**Video 12.** Simulation of *axn* RNAi and hs-hb; *axn* RNAi phenotypes in *Tribolium*. Shown are simulations of the SR model in simulated (**A**) *axn* RNAi background and (**B**) hs-hb; *axn* RNAi in *Tribolium*. The *axn* RNAi phenotype is simulated by not retracting the *cad* gradient (grey).
DOI: https://doi.org/10.7554/eLife.41208.031

clock can be regulated by the relative proportions of the dynamic and static modules. In the FFTG model, the clock is the Timer Gene, where each tick of the clock is specified by a different concentration of the Timer Gene. Different concentrations of the Timer Gene are translated into different fates by means of a FF model. The speed of the Timer Gene clock is regulated by the activating posterior morphogen (grey in *Figure 2C*), since the concentration of an activator naturally regulates the transcription rate of its target gene. Although a gene with expression dynamics like that of a Timer Gene has not so far been discovered in insects, its existence is still a viable possibility.

Resetting the clock of the SR model would, then, requireresetting the gap gene cascade itself, while resetting the clock of the FFTG model would require resetting the Timer Gene. To investigate whether gap gene regulation in insects are mediated by the SR or the FFTG

model, we re-induced the leading gap gene in the gap gene sequence (*hb*) in the *Tribolium* embryo, which resulted in the re-induction of the gap gene cascade. These experimental observations are consistent with a Speed Regulation model of gap gene regulation in *Tribolium*. Interestingly, hox genes are expressed sequentially in waves in vertebrates, and are hypothesized to be regulated by a clock mechanism (termed the 'hox clock') (*Deschamps and Duboule, 2017*; *Deschamps and van Nes, 2005*; *Gaunt, 2001*). It is not clear, however, if this clock is self-regulatory or regulated by a threshold-based mechanism. Similar perturbations like the one suggested in this study might resolve this issue.

One limitation of our theoretical analysis and comparison of the SR, FF, and FFTG models is that we used specific GRN realizations for these models, since other GRN realizations could also be possible. However, we used our GRN realizations to illustrate arguments that are general. The GRN realizations of FF and FFTG models are used to show that important classes of FF realizations exhibit similar dynamics to the SR model. The failure of FF GRN to re-induce the patterning genes sequence upon the re-induction of one of the constituent genes is due to a general feature of the FF model rather than to some modeling specifics of the used GRNs, namely due to that fact that the morphogen gradient is the driving factor behind the sequential activation of genes rather than the cross-regulatory interactions between patterning genes themselves, as the case in the SR model.

In this paper, we used a GRN realization of the SR model based on the gradual switching between two genetic circuits. Another realization of the SR model would be to regulate both the activation and decay rates of the constituent genes of a genetic cascade by the morphogen gradient. However, the dual response of *Tribolium* embryos upon re-inducing *hb* supports the two-module realization (compare *Videos 8* and *11*). Yet another possible GRN realization for the SR model is the composite AC/DC mode of the AC-DC motif (*Perez-Carrasco et al., 2018*). The AC-DC GRN can operate either as a multi-stable circuit (the DC mode) or as a clock (the AC mode), depending on the concentration of a morphogen gradient. At certain ranges of the morphogen gradient, both modes could co-exist, resulting in the fine-tuning of the speed of the clock. Hence, the AC-DC GRN has a somewhat similar logic as the two-modules GRNs; however, it uses one gene circuit that can work as both dynamic or static modules depending on an external factor rather than explicitly using two genetic modules as in the two-modules GRN. However, it is not clear if the AC-DC GRN would be able to mediate a morphogen-mediated transition from the composite AC-DC mode to a pure DC mode to fully realize wavefront-based patterning. In addition, while the AC-CD GRN is economical as it uses one module to achieve both AC and DC behaviors, it lacks the flexibility of the two-modules model, since different realizations of the dynamic and static modules in the two-modules model could be employed to mediate a wide range of final spatial patterns without the constraint of using one module to mediate two functions.

In this paper, we studied the effect of re-inducing *hb* on gap gene regulation. However, gap genes are known to interact with pair-rule genes (albeit a limited interaction in *Tribolium* (*El-Sherif et al., 2012b*)). Indeed, *hb* re-activation in our experiments induced the generation of extra segments in cuticles (see Appendix 1 for a basic description of cuticlular phenotype of hs-hb experiments). Interestingly, depleting *hb* transcripts by RNAi led to extra segments as well in the moth *Bombyx mori* (*Nakao, 2016*) (a phenotype that is not observed upon knocking down *hb* in *Tribolium*). However, this could be either due to a difference in the mode of gap gene regulation between the two species or due to the timing of applying RNAi perturbation (as the *hb* RNAi was done via embryonic injection in *Bombyx* and via parental injection in *Tribolium*, leading to the depletion of both maternal and zygotic *hb*). Investigating this and the interaction between gap and pair-rule gene networks in general will be addressed in future work.

In summary, in this paper we showed that the gene expression dynamics driven by a French Flag mechanism can be indistinguishable from those driven by a threshold-free mechanism. We then suggested a test to differentiate between the two cases and carried it out experimentally in the beetle *Tribolium castaneum*. The test confirmed that the AP fate specification in *Tribolium* (and possibly other insects) is based on the regulation of an aperiodic clock of gap genes in a threshold-free fashion.

## Materials and methods

**Key resources table**

| Reagent type (species) or resource | Designation | Source or reference | Identifiers | Additional information |
|---|---|---|---|---|
| Strain (*Tribolium castaneum*) | San Bernardino WT (SB) | NA | NA | Most *Tribolium* labs (including El-Sherif and Klinger labs) |
| Strain background (*Tribolium castaneum*) | vermilion white v(w) | NA | NA | Most *Tribolium* labs (including El-Sherif and Klinger labs) |
| Recombinant DNA reagent | pBac[3xP3-v; Tc'hsp68-Tc'hb-Tc'hsp68 3'UTR] | NA | NA | Klingler lab |

### In situ hybridization, RNAi, and imaging of fixed embryos

In situ hybridization was performed using DIG-labeled RNA probes and anti-DIG::AP antibody (Roche). Signal was developed using NBT/BCIP (BM Purple, Roche) according to standard protocols (*Schinko et al., 2009*; *Shippy et al., 2009*). All expression analyses were performed using embryos from uninjected females or females injected with double-stranded RNA (dsRNA) of gene of interest. dsRNA was synthesized using the T7 megascript kit (Ambion) and mixed with injection buffer (5 mM KCl, 0.1 mM KPO$_4$, pH 6.8) before injection. Used dsRNA concentration for *axn* RNAi: 100 ng/µl. Embryos were imaged using ProgRes CFcool camera on Zeiss Axio Scope.A1 microscope and ProgRes CapturePro image acquisition software. Brightness and contrast of all images were adjusted and placed on a white background using Adobe Photoshop.

### Overexpression construct

A 1757 bp fragment of the *Tc-hb* mRNA cDNA22 (*Wolff et al., 1995*) was amplified using PCR primers *Tc-hb*_left 5'-CGTCTAGAGCAAAAATTTCGAACAGTCG-3' and *Tc-hb*_right 5'-CCGCTCGAG TCCAACCCGTACATCTCCAT-3' which were designed to include restriction sites XbaI and XhoI, respectively. This *Tc-hb* fragment contains the complete coding region and partial 5'UTR and 3'UTR sequences and was cloned between a *Tc'hsp68* promotor and 3'UTR sequences via the XbaI-XhoI sites in plasmid pSLfa[Tc'hsp5'-dsRedEx-3'UTR; 5'3'UTR]fa (Johannes Schinko and Gregor Bucher, unpublished). From this plasmid, an AscI-FseI fragment including the hs-hb cassette was subcloned as AscI-FseI fragment into the piggyBac transformation vector pBac[3xP3-gTcv] (Johannes Schinko, unpublished), resulting in pBac[3xP3-v; Tc'hsp68-Tc'hb-Tc'hsp68 3'UTR] (*Figure 4—figure supplement 1*). This vector uses the *Tribolium vermilion* gene as transformation marker (*Lorenzen et al., 2002*).

### Generation of hs-hb transgenic beetles

Plasmid DNAs were isolated using the Quiagen plasmid Midi Kit, and germline transformation was performed as described in refs (*Berghammer et al., 1999*; *Berghammer et al., 2009*). In one experimental series, 408 vermilion white embryos were injected of which 22% hatched. 44 crosses were set up, from which 10 transgenic strains could be generated. In another experimental series, 210 embryos were injected with a hatch rate of 56%. 59 crosses were set up, from which five transgenic strains could be generated. Ten of these hs-hb lines were tested for heat-shock phenotypes. Phenotype strength was measured by determining the proportion of larvae which (*i*) developed homeotic transformations, and (*ii*) which displayed, in addition to the homeotic transformations, additional trunk segments (see Appendix 1 for basic description of the cuticle phenotype of heat-shocked hs-hb embryos). Two out of those ten lines (hs-hb one and hs-hb 2) seemed most effective in generating heat-shock phenotypes and were further studied. The strain hs-hb two was used to generate the data in this paper.

### Non-heat-shocked egg collections

Three hours developmental windows were generated by incubating three-hours egg collections at 23–24°C for the desired length of time before fixation. Beetles were reared in whole-wheat flour supplemented with 5% dried yeast.

## Heat-shocked egg collections

Three hours developmental windows were generated by incubating three-hours egg collections at 23–24°C for the desired length of time. Egg collections are then heat-shocked in a water bath at 48°C for 10 min and then re-incubated at 23–24°C for the desired length of time before fixation.

## Quantification of gene expressions in HS Anterior, Anterior, and Active-Zone

Gene expression quantifications (*Figure 4C* and *Figure 5*, with numerical data in *Figures 4—figure supplement 3*, *Figure 5—figure supplements 2* and *4*, *Figure 5—figure supplement 4* see source data in *Figure 5—source data 1*, and *Figure 5—source data 2*) were created by counting proportions of embryos that have detectable expression in the three regions: HS Anterior, Anterior, and Active-Zone. Dividing an embryo into HS Anterior, Anterior, and Active-Zone is carried out using morphological markers in the *Tribolium* germband (*Figure 4B*) as follows. The posterior end of the germband usually has a roundish shape that gets gradually fused into a long rectangular shape as we go towards anterior, ending with the head at far anterior. The 'Active-Zone' starts from the posterior end of the germband and ends at the point of fusion of the roundish posterior and the rectangular region of the embryo. The 'Anterior' is everything anterior to the Active-Zone. The 'HS Anterior' is only the rectangular region of the germband. These morphological landmarks are still present in heat-shocked hs-hb germbands, albeit the whole germband is shortened.

Error bars in *Figures 4—figure supplement 3*, *Figure 5—figure supplements 2* and *4*, Figures 4-figure supplement 4 represent standard error (*SE*) of proportions, estimated with the formula:

$$SE = \sqrt{\frac{p(1-p)}{n}}$$

where $p$ is the proportion of embryos with detectable expression in the designated region of the embryo (either HS Anterior, Anterior, or Active Zone) within an egg collection, while $n$ is the total number of embryos in the egg collection.

We used egg collections of 15 embryos on average (see *Figure 5—source data 1*, *Figure 5—source data 1*, and *Figure 5—source data 2* for exact sample sizes), which yielded standard errors small enough for our analysis (See *Figures 4—figure supplement 3*, *Figure 5—figure supplements 2* and *4*, Figures 4-figure supplement 4). We used one replicate for each egg collection (per gene visualized per time point). However, the large number of consecutive time points and parallel egg collections (each per gene visualized) carried out in this study confirms the described trend in the presented data. A total of 3500 embryos were analyzed in this study.

## Computational modeling

See Appendix 1 and *Supplementary file 1*.

# Acknowledgements

We thank Paul François and Erik Clark for their comments on an early version of the manuscript.

# Additional information

## Funding

| Funder | Grant reference number | Author |
| --- | --- | --- |
| Alexander von Humboldt-Stiftung | Fellowship | Ezzat El-Sherif |
| Deutsche Forschungsgemeinschaft | KL 656_5-1 | Martin Klingler |

The funders had no role in study design, data collection and interpretation, or the decision to submit the work for publication.

## Author contributions
Alena Boos, Jutta Distler, Heike Rudolf, Investigation; Martin Klingler, Conceptualization, Funding acquisition, Investigation; Ezzat El-Sherif, Conceptualization, Supervision, Funding acquisition, Investigation, Writing—original draft

## Author ORCIDs
Ezzat El-Sherif (iD) http://orcid.org/0000-0003-1738-8139

## Decision letter and Author response
Decision letter https://doi.org/10.7554/eLife.41208.036
Author response https://doi.org/10.7554/eLife.41208.037

# Additional files

## Supplementary files
• Supplementary file 1. Matlab implementation of models and simulations. Matlab implementations for the different models presented in our study and the parameter sets used to generate each of our simulations (*Videos 1–12*).
DOI: https://doi.org/10.7554/eLife.41208.032

• Transparent reporting form
DOI: https://doi.org/10.7554/eLife.41208.033

## Data availability
Numerical data and sample sizes are all documented in Figure 4–source data 1, Figure 5–source data 1, and Figure 5–source data 2

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

# Appendix 1

DOI: https://doi.org/10.7554/eLife.41208.034

## Cuticular and morphological phenotype of heat-shocked hs-hb embryos

Hunchback overexpressing larvae show two main phenotypic traits, homeotic transformation of one or several (sometimes most) abdominal segments into segments with thoracic identity. Often imperfect transformations are observed, for example development of incomplete leg structures, or occurrence of fully transformed segments surrounded by segments with weak transformation. A subset of affected larvae display, in addition to the homeotic transformation, a disturbed segment pattern in the abdomen, sometimes with several fused or partially fused abdominal segments. Most interestingly, a sizable portion of larvae with homeotic transformation display supernumerary abdominal segments such that up to a total number of 19 trunk segments might be found. Germbands of heat-shocked hs-hb embryos appear shorter than WT embryos at the same stage, either due to the involvement of gap genes directly in the process of convergent extension or due to defects in the segmentation process, as segmentation seems to direct convergent extension in *Tribolium* (**Benton et al., 2016**).

## A basic description of how French Flag GRN works

In the example GRN in **Figure 1A''**, the morphogen gradient (grey) activates different genes with different strengths: strongly activates the blue gene, moderately activates the red gene, and weakly activates the green gene. Being strongly activated by the morphogen gradient, the blue gene is first activated in all cells exposed to the morphogen gradient. Being moderately activated by the morphogen, the red gene is then (after a delay) activated only at regions where the morphogen value exceeds its activation threshold. Since the red gene represses the blue gene, the blue gene now turns off where the red gene is now expressed. In a similar fashion, the green gene turns on (after a delay) only in regions where the morphogen concentration exceeds its activation threshold and then, eventually, represses the red gene there.

## Computational Modeling

All computational models were created using Matlab. For all GRN models in this study, the transcriptional activity $E$ of gene $X$ that is regulated by $N_a$ activators ($A_i$, $i$ = 1 to $N_a$) and $N_r$ repressors ($R_j$, $j$ = 1 to $N_r$) is modelled using AND gated Hill functions of individual binding proteins:

$$E = \prod_{i=1}^{Na} \frac{(A_i/K_i)^{na_i}}{1+(A_i/K_i)^{na_i}} \prod_{j=1}^{Nr} \frac{1}{1+\left(R_j/K_j'\right)^{nr_j}}$$

where $A_i$ is the concentration of activator $A_i$, $R_j$ is the concentration of repressor $R_j$, $na_i$ is the cooperativity of $A_i$, $nr_j$ is the cooperativity of $R_j$, $K_i$ is the dissociation constant of activator $A_i$, and $K_j'$ is the dissociation constant of repressor $R_j$.

Below are complete differential equations for GRN models used in this study.

## French Flag GRN

The following are the equations used for modeling 5-genes French Flag GRN (5-genes version of the GRN in **Figure 1A''**). $X_i$ is the mRNA concentration transcribed by gene $X_i$. $G$ is the concentration of the gradient $G$ (see Gradient Setup section below). Autorepression links are added for steady-state level control.

$$\frac{dX_1}{dt} = \frac{\left(\frac{G}{0.2}\right)^3}{1+\left(\frac{G}{0.2}\right)^3} \times \frac{1}{1+\left(\frac{X_2}{2.5}\right)^5} \times \frac{1}{1+\left(\frac{X_3}{0.2}\right)^5} \times \frac{1}{1+\left(\frac{X_4}{0.2}\right)^5} \times \frac{1}{1+\left(\frac{X_5}{0.2}\right)^5} \times \frac{1}{1+\left(\frac{X_1}{0.8}\right)^5} - X_1$$

$$\frac{dX_2}{dt} = \frac{\left(\frac{G}{0.6}\right)^3}{1+\left(\frac{G}{0.6}\right)^3} \times \frac{1}{1+\left(\frac{X_3}{2.5}\right)^5} \times \frac{1}{1+\left(\frac{X_4}{0.2}\right)^5} \times \frac{1}{1+\left(\frac{X_5}{0.2}\right)^5} \times \frac{1}{1+\left(\frac{X_2}{0.8}\right)^5} - X_2$$

$$\frac{dX_3}{dt} = \frac{\left(\frac{G}{1}\right)^3}{1+\left(\frac{G}{1}\right)^3} \times \frac{1}{1+\left(\frac{X_4}{2.5}\right)^5} \times \frac{1}{1+\left(\frac{X_5}{0.2}\right)^5} \times \frac{1}{1+\left(\frac{X_3}{0.8}\right)^5} - X_3$$

$$\frac{dX_4}{dt} = \frac{\left(\frac{G}{1.4}\right)^3}{1+\left(\frac{G}{1.4}\right)^3} \times \frac{1}{1+\left(\frac{X_5}{0.4}\right)^5} \times \frac{1}{1+\left(\frac{X_4}{0.8}\right)^5} - X_4$$

$$\frac{dX_5}{dt} = \frac{\left(\frac{G}{1.8}\right)^3}{1+\left(\frac{G}{1.8}\right)^3} \times \frac{1}{1+\left(\frac{X_5}{0.8}\right)^5} - X_5$$

Initial conditions: $X_1=X_2=X_3=X_4=X_5=0$.

## Speed Regulation GRN (Module Switching model)

The transcription rate of gene $X$ ($X$) is modeled as a weighted sum of the activity of two modules, dynamic module ($X_D$) and static module ($X_s$),

$$X = c_1 X_D + c_2 X_S$$

The following are the equations used for modeling the 5-genes gradual module switching model (5-genes version of the GRN in *Figure 1B''*). $X_{Di}$ is the activity of the dynamic module of gene $X_i$. $X_{Si}$ is the activity of the static module of gene $X_i$. $X_i$ is the mRNA concentration transcribed by gene $X_i$. $G$ is the concentration of the (speed regulator) gradient $G$ (see 'Modeling gradients and gradient dynamics' section below).

Dynamic Modules:

$$\frac{dX_{D1}}{dt} = \frac{G}{1+G} \times \frac{1}{1+\left(\frac{X_2}{0.4}\right)^5} \times \frac{1}{1+\left(\frac{X_3}{0.4}\right)^5} \times \frac{1}{1+\left(\frac{X_4}{0.4}\right)^5} \times \frac{1}{1+\left(\frac{X_5}{0.4}\right)^5}$$

$$\frac{dX_{D2}}{dt} = \frac{G}{1+G} \times \frac{1}{1+\left(\frac{X_1}{2.5}\right)^5} \times \frac{1}{1+\left(\frac{X_3}{0.4}\right)^5} \times \frac{1}{1+\left(\frac{X_3}{0.4}\right)^5} \times \frac{1}{1+\left(\frac{X_5}{0.4}\right)^5}$$

$$\frac{dX_{D3}}{dt} = \frac{G}{1+G} \times \frac{1}{1+\left(\frac{X_1}{0.4}\right)^5} \times \frac{1}{1+\left(\frac{X_2}{2.5}\right)^5} \times \frac{1}{1+\left(\frac{X_4}{0.4}\right)^5} \times \frac{1}{1+\left(\frac{X_5}{0.4}\right)^5}$$

$$\frac{dX_{D4}}{dt} = \frac{G}{1+G} \times \frac{1}{1+\left(\frac{X_1}{0.4}\right)^5} \times \frac{1}{1+\left(\frac{X_2}{0.4}\right)^5} \times \frac{1}{1+\left(\frac{X_3}{2.5}\right)^5} \times \frac{1}{1+\left(\frac{X_5}{0.4}\right)^5}$$

$$\frac{dX_{D5}}{dt} = \frac{G}{1+G} \times \frac{1}{1+\left(\frac{X_1}{0.4}\right)^5} \times \frac{1}{1+\left(\frac{X_2}{0.4}\right)^5} \times \frac{1}{1+\left(\frac{X_3}{0.4}\right)^5} \times \frac{1}{1+\left(\frac{X_4}{2.5}\right)^5}$$

Static Modules:

$$\frac{dX_{S1}}{dt} = \frac{1}{1+G} \times \frac{1}{1+\left(\frac{X_2}{0.4}\right)^5} \times \frac{1}{1+\left(\frac{X_3}{0.4}\right)^5} \times \frac{1}{1+\left(\frac{X_4}{0.4}\right)^5} \times \frac{1}{1+\left(\frac{X_5}{0.4}\right)^5}$$

$$\frac{dX_{S2}}{dt} = \frac{1}{1+G} \times \frac{1}{1+\left(\frac{X_1}{0.4}\right)^5} \times \frac{1}{1+\left(\frac{X_3}{0.4}\right)^5} \times \frac{1}{1+\left(\frac{X_4}{0.4}\right)^5} \times \frac{1}{1+\left(\frac{X_5}{0.4}\right)^5}$$

$$\frac{dX_{S3}}{dt} = \frac{1}{1+G} \times \frac{1}{1+\left(\frac{X_1}{0.4}\right)^5} \times \frac{1}{1+\left(\frac{X_2}{0.4}\right)^5} \times \frac{1}{1+\left(\frac{X_4}{0.4}\right)^5} \times \frac{1}{1+\left(\frac{X_5}{0.4}\right)^5}$$

$$\frac{dX_{S4}}{dt} = \frac{1}{1+G} \times \frac{1}{1+\left(\frac{X_1}{0.4}\right)^5} \times \frac{1}{1+\left(\frac{X_2}{0.4}\right)^5} \times \frac{1}{1+\left(\frac{X_3}{0.4}\right)^5} \times \frac{1}{1+\left(\frac{X_5}{0.4}\right)^5}$$

$$\frac{dX_{S5}}{dt} = \frac{1}{1+G} \times \frac{1}{1+\left(\frac{X_1}{0.4}\right)^5} \times \frac{1}{1+\left(\frac{X_2}{0.4}\right)^5} \times \frac{1}{1+\left(\frac{X_3}{0.4}\right)^5} \times \frac{1}{1+\left(\frac{X_4}{0.4}\right)^5}$$

Combining the activities of Dynamic and Static Modules:

$$\frac{dX_1}{dt} = 3\frac{dX_{D1}}{dt} + 2\frac{dX_{S1}}{dt} - X_1$$

$$\frac{dX_2}{dt} = 3\frac{dX_{D2}}{dt} + 2\frac{dX_{S2}}{dt} - X_2$$

$$\frac{dX_3}{dt} = 3\frac{dX_{D3}}{dt} + 2\frac{dX_{S3}}{dt} - X_3$$

$$\frac{dX_4}{dt} = 3\frac{dX_{D4}}{dt} + 2\frac{dX_{S4}}{dt} - X_4$$

$$\frac{dX_5}{dt} = 3\frac{dX_{D5}}{dt} + 2\frac{dX_{S5}}{dt} - X_5$$

Initial conditions: $X_1 = 0.1$, $X_2 = X_3 = X_4 = X_5 = 0$.

We would like to note here that in this paper, we used the 'module switching model' (Zhu et al PNAS 2017) as a molecular realization of the speed regulation model. The model basically posits that gene regulation changes its wiring depending on the presence of a posterior morphogen or not. This prediction is supported by the dual response of the *Tribolium* embryo upon re-inducing *hb* reported in this paper. The 'module switching' can be realized either by assuming that the same enhancers change their wiring depending on posterior morphogen concentration, or, alternatively, there is a set of enhancers that are active in the presence of the morphogen and other set of enhancers active in the absence of the morphogen. In Zhu et al PNAS 2017, we favored the latter possibility. However, the 'two enhancer sets' assumption is not an essential part of the 'module switching' model.

## French Flag with a Timer Gene GRN

Same as French Flag GRN but with gradient *G* replaced with a Timer Gene *TG* as activator of genes $X_i$, $i$ = 1 to 5. *TG* is activated by the gradient *G* and has zero decay rate:

$$\frac{dTG}{dt} = 0.05 \times G$$

## Speed Regulation model realization by jointly modulating gene activation and gene product decay rates

Another realization of the Speed Regulation model is for the morphogen gradient to activate the constituent genes of a genetic cascade (or oscillator) and at the same time positively regulate their decay rates. The following is a set of differential equations realizing this concept. $X_i$ is the mRNA concentration transcribed by gene $X_i$. *G* is the concentration of the (speed regulator) gradient **G**.

$$\frac{dX_1}{dt} = \frac{G}{1+G}\left(\frac{1}{1+\left(\frac{X_2}{0.4}\right)^5} \times \frac{1}{1+\left(\frac{X_3}{0.4}\right)^5} \times \frac{1}{1+\left(\frac{X_4}{0.4}\right)^5} \times \frac{1}{1+\left(\frac{X_5}{0.4}\right)^5} - 3X_1\right)$$

$$\frac{dX_2}{dt} = \frac{G}{1+G}\left(\frac{1}{1+\left(\frac{X_1}{2.5}\right)^5} \times \frac{1}{1+\left(\frac{X_3}{0.4}\right)^5} \times \frac{1}{1+\left(\frac{X_4}{0.4}\right)^5} \times \frac{1}{1+\left(\frac{X_5}{0.4}\right)^5} - 3X_2\right)$$

$$\frac{dX_3}{dt} = 2 \times \frac{G}{1+G}\left(\frac{1}{1+\left(\frac{X_1}{0.4}\right)^5} \times \frac{1}{1+\left(\frac{X_2}{2.5}\right)^5} \times \frac{1}{1+\left(\frac{X_4}{0.4}\right)^5} \times \frac{1}{1+\left(\frac{X_5}{0.4}\right)^5} - 3X_3\right)$$

$$\frac{dX_4}{dt} = 2 \times \frac{G}{1+G}\left(\frac{1}{1+\left(\frac{X_1}{0.4}\right)^5} \times \frac{1}{1+\left(\frac{X_2}{0.4}\right)^5} \times \frac{1}{1+\left(\frac{X_3}{2.5}\right)^5} \times \frac{1}{1+\left(\frac{X_5}{0.4}\right)^5} - 3X_4\right)$$

$$\frac{dX_5}{dt} = 2 \times \frac{G}{1+G}\left(\frac{1}{1+\left(\frac{X_1}{0.4}\right)^5} \times \frac{1}{1+\left(\frac{X_2}{0.4}\right)^5} \times \frac{1}{1+\left(\frac{X_3}{0.4}\right)^5} \times \frac{1}{1+\left(\frac{X_4}{2.5}\right)^5} - 3X_5\right)$$

## Modeling gradients and gradient dynamics

Here we present how we modeled gradients with different dynamics in our simulations. Consider a group of cells arranged along a spatial axis $x$. We then apply a concentration gradient $G$ along $x$. The gradient could be of two forms: (*i*) a smooth non-retracting gradient, and (*ii*) a retracting steep gradient (i.e. a retracting boundary or a wavefront). The smooth static gradient resembles that of *caudal* (*cad*; a strong candidate for the speed gradient in *Tribolium*) during the blastoderm stage of insect embryogenesis. The retracting wavefront is analogous to *cad* expression during the germband stage.

Here we will devise two mathematical formulae for each of the two forms of the speed gradient.

The smooth static gradient can be modeled with the sigmoid function:

$$G(x) = \frac{1}{1+e^{-m(x-a)}}$$

The infliction point of the sigmoid is specified by $a$. The constant $m$ specifies how steep the sigmoid is. A value of $m = 20$ gives reasonably smooth gradient and matches well the expression of *cad* in the blastoderm of the intermediate germ insect *Tribolium castaneum*.

To model a retracting wavefront with speed $v$, the following modified version of the sigmoid function can be used:

$$G(t,x) = \frac{1}{1+e^{-m(x-a-vt)}}$$

We find a value of $m = 100$ to yield a reasonably steep wavefront.

Since in most insects, the blastoderm stage eventually transits into a germband stages, we will devise a flexible mathematical formula for the gradient so that it can (smoothly) transit from the static smooth form to the retracting wavefront form. If the transition from the blastoderm to germband takes place at $t = T_{bg}$, then $G$ could be written as follows:

$$G(t,x) = \frac{1}{1+e^{-m(t)(x-a-u(t)t)}}$$

where,

$$m(t) = \begin{cases} 20 \,, & t < ; T_{bg} \\ 20e^{10\left(t-T_{bg}\right)}, & t \geq T_{bg} \end{cases}$$

and,

$$u(t) = \begin{cases} 0 \ , \ t <; T_{bg} \\ v, \ t \geq T_{bg} \end{cases}$$

In the case of a static gradient, we model the dynamics of gradient buildup or decay by multiplying the static gradient formula $G$ mentioned above by a dynamics term $D(t)$, where

$$D(t) = 1 - \left( \alpha \frac{t}{T} - \beta \right)^2$$

and,

$$G(t,x) = D(t) \frac{1}{1 + e^{-m(x-a-vt)}}$$

Parameters $\alpha$ and $\beta$ specifies what type of dynamics $D(t)$ introduces. For $\alpha = 1$ and $\beta = 1$, $D(t)$ will introduce buildup dynamics. For $\alpha = 2$ and $\beta = 1$, $D(t)$ will introduce buildup followed by decay dynamics. For $\alpha = 1$ and $\beta = 0$, $D(t)$ will introduce only decay dynamics. $T$ is the total duration of the simulation.

## Modeling heat shock experiments

Heat-shocked hs-hb experiments were simply modeled by introducing a pulse of transcription in the anterior most expressed gene ($X_1$) at a desired point of time. Let $\frac{dX_1}{dt}$ be the rate of transcription of X1 without introducing the possibility of applying a heat-shock experiment (like the formulae of $\frac{dX_1}{dt}$ introduced in different models of patterning above). Then the rate of transcription of $X_1$ after introducing the possibility of applying a heat-shock experiment ($X'_1$) will be:

$$\frac{dX'_1}{dt} = H(t) + \frac{dX_1}{dt}$$

where the heat-shock term $H(t)$ is given by,

$$H(t) = \begin{cases} h \ , \ Ths_1 < t < Ths_2 \\ 0, \ \text{elsewhere} \end{cases}$$

where $Ths_1$ and $Ths_2$ are start and end times of heat-shock application, and $h$ is the transcription rate of the heat-shock transgene under the heat-shock conditions.

## Modeling the *axin* phenotype

We modeled the *axin* phenotype using Speed Regulation model of patterning by simply not retracting the posterior gradient (speed regulator). This is equivalent to our simulation of the long-germ mode of embryogenesis (**Video 6C** and **Video 12A**). In **Video 12B**, we applied the heat-shock perturbation to this model.

## Matlab implementation

Matlab implementations for the different models presented in our study and the parameter sets used to generate each of our simulations (**Videos 1–12**) can be found in **Supplementary file 1**.

