## [Decision Letter]

Thank you for submitting your article "A re-inducible gap gene cascade patterns the anterior-posterior axis of insects in a threshold-free fashion" for consideration by *eLife*. Your article has been reviewed by two peer reviewers, and the evaluation has been overseen by Naama Barkai as the Senior and Reviewing Editor. The reviewers have opted to remain anonymous.

The reviewers have discussed the reviews with one another and the Reviewing Editor has drafted this decision to help you prepare a revised submission.

Summary:

Boos et al. propose that the anterior-posterior axis in Tribolium is regulated by a threshold-free speed regulation mechanism based on a gradient of caudal and gap gene interactions. This is largely based on the observation that pulsed hb expression re-initiates the patterning cascade, which is roughly consistent with the predictions from a model of the gap gene network.

Essential revisions:

1) Include a basic description of the HS-hb phenotype (or range of phenotypes) – i.e. describe the effects on segment number, segment identity, duration of segmentation, and germband morphology. The data visualization in Figures 4—figure supplement 3, Figure 5—figure supplement 2 and Figure 5—figure supplement 4 should also be improved.

2) The paper should be rewritten to minimize content overlap with the PNAS paper, i.e. streamline everything before "A dual response upon re-inducing hunchback in the Tribolium embryo", and appropriately discuss previous studies.

3) Address the axin RNAi results with additional simulations (i.e. show the simulation results for the axin RNAi experiment), and examine gap gene expression after cad RNAi (using late stage knockdown if appropriate).

4) Add missing details regarding the methodology (e.g. dsRNA concentration, gradient modeling, morphological landmarks to define the three embryonic regions, etc..)

Strongly encouraged revisions:

1) Repeat the hs:hb experiment in a caudal loss-of-function background (mutant or RNAi) to assess whether patterning after hb activation is still dependent on the initial morphogen caudal.

2) Provide examples of enhancers in the gap gene network, in which caudal acts as both an activator and a repressor for a given target gene to make the work less speculative.

Additional desirable revisions:

1) Optimize the heatshock conditions to make the experiments interpretable in the framework of the speed-regulation model.

2) To add novelty to the model, the paper could benefit from an extension to analyze dynamic gradient formation rather than assuming a static caudal protein gradient for the initial phases of patterning.

3) Tone down the relevance for patterning in other insects unless additional evidence is provided. It would be interesting for the authors to comment on the relationships between their own findings and those of Hajime Nakao in the Bombyx embryo (Nakao 2016). That study implicated Hb in the termination of segment addition and Hox gene expression, rather than reinitiating it.

In addition, please consider all points in the individual reviews below.

*Reviewer #1:*

Most bilaterians generate their main body axis progressively over time, from anterior to posterior. During axial extension, expression domains of axial patterning genes (e.g. Hox genes, or the gap genes as discussed here) appear dynamically and sequentially in posterior, unspecified tissue; this temporal pattern is the origin of the spatial pattern of axial identity seen in the fully-developed animal. In general, two contrasting regulatory mechanisms might give rise to the temporal pattern. One option is that residence time within the progenitor zone is measured directly, on the basis of some "timer" signal that increases or decreases as the zone matures. Alternatively, the patterning genes themselves might constitute the timer mechanism, with regulatory interactions between the various genes resulting in dynamic and sequential expression. In the first case, the patterning genes simply respond to temporal information, in the second case, they create it.

Boos and colleagues aim to distinguish between these possibilities in the case of a series of gap genes expressed in a sequentially segmenting insect, Tribolium castaneum. These genes are expressed in a temporal sequence (hb, Kr, mlpt, gt), and contribute to axial patterning of the embryo. A previous paper from the authors proposed that these genes form a genetic cascade, and that the rate of progression through this cascade is regulated in a concentration dependent manner by a posteriorly expressed factor, Caudal. This model is an example of the latter category of timer mechanisms. As a representative of the former category, the authors point to a model proposed by Francois and Siggia in 2010, whereby the posterior zone promotes the expression of a timer gene, which successively activates more "posterior" gap genes as its expression level progressively builds up over time. (In that paper, Caudal itself was identified as a potential timer gene, but Caudal levels have not been shown to increase over time in the manner suggested. Instead, in the formulation in the current manuscript, Caudal activates a hypothetical and as yet undiscovered timer gene.)

The authors find that misexpressing hunchback (the first gene in the gap sequence) using a heat-shock promoter turns out to reinitiate the entire four gene sequence of gap gene expression, as predicted by the genetic cascade model, and against the predictions of the "timer gene" alternative. Experiments comparing HS-Hb with HS-Hb; axn RNAi embryos further show that the zone of re-initiation consistently correlates with the extent of caudal expression, indicating that genetic regulatory logic differs between different regions of the embryo, and that Caudal may be (one of) the factors responsible for this difference.

The HS-Hb treatment is a neat functional perturbation, while the experimental findings of gap gene expression re-initiation are striking, and the theoretical predictions for the genetic cascade vs timer gene scenarios are well contrasted. As such, the manuscript supports and extends existing ideas in the field about the role of gap gene networks in insect development. However, I have reservations about the focus, organisation and presentation of the paper, which could all be significantly improved.

1) The manuscript takes a long time (nearly 200 lines) to home in on the question actually addressed by the experiments – that of a genetic cascade vs a concentration dependent timer gene. I think that readability, focus, and clarity would all be improved by streamlining the paper and removing/drastically shortening the long introductory diversion into models of patterning mediated by static morphogen gradients. Aside from being beside the point, much of this material simply repeats the authors' 2017 PNAS paper, or re-describes points already made in Francois and Siggia 2010. (Indeed, Figures 1B, 1D, 2A, 2B, and 3A all reproduce images from the 2017 paper, without remark.) In addition, I am concerned about a misleading presentation of *Drosophila* gap gene regulation as being of the "French Flag" type, instead of discussing the authors' models in the context of published work on emergent gap gene dynamics from the Reinitz, Jaeger, and Gregor labs.

2) I find some of the terminology used when discussing the models to be unhelpful – particular the comparison between "French flag"/"threshold-based" scenarios and the authors' "speed-regulation" model. First, the key hypothesis being tested is the presence of a concentration dependent timing factor in the "active" zone, vs timing information emerging from the interplay of the gap factors themselves. (I have used the terms "timer gene" and "genetic cascade" in this review to refer to these respective concepts.) While the authors repeatedly state that their findings validate their "speed regulation" model (e.g. subsection “A dual response upon re-inducing hunchback in the Tribolium embryo”, fifth paragraph, Discussion, third and last paragraphs), this claim is potentially misleading as they instead simply rule out the "timer gene" hypothesis; the experimental results are not informative regarding the "speed regulation" aspect of the speed regulation model.

Second, the use of "threshold-free" to refer to the absence of a concentration-dependent timer gene is potentially confusing, because this "threshold-free" scenario doesn't rule out the existence of other regulatory thresholds within the genetic network. Third, the authors use the French flag model as a reference point when introducing their models. However, they also argue for a redefinition of the French flag (Introduction, second paragraph) that fits with their GRN implementations. Given that their study system isn't well represented by a French flag scenario anyway, this strategy seems unnecessarily complicated and potentially confusing.

3) While the paper is understandably focused on the effects on gap gene expression, it should also include some basic description of the terminal phenotypes resulting from the HS-Hb treatments. There is a line in the Materials and methods section (subsection “Generation of transgenic beetles”) about larvae developing homeotic transformations and/or additional trunk segments. This strikes me as very interesting (and increases support for the authors' model) and I would have ideally liked to have seen some characterisation of these effects: for example, cuticle preparations and in situs against Hox genes and segmentation genes. In addition, the HS-Hb germbands shown in Figure 4—figure supplement 2, Figure 5—figure supplement 1, and Figure 5—figure supplement 1 are clearly much shorter than their HS-wt counterparts (whereas the model suggests that they ought to generate more segments). This is also interesting but not remarked upon. Is the morphological difference due to effects on convergent extension, cell proliferation, or both? And, if these short germbands eventually produce larvae with extra segments, is the duration of segmentation/axial extension therefore significantly protracted?

*Reviewer #2:*

In this manuscript, Boos et al. propose that the anterior-posterior axis in Tribolium is regulated by a threshold-free speed regulation mechanism based on a gradient of caudal and gap gene interactions. This is largely based on the observation that pulsed hb expression re-initiates the patterning cascade, which is roughly consistent with the predictions from a model of the gap gene network. The experimental observation is very interesting, but the model in its current state remains speculative. The study therefore needs to be extended to further test the validity of the model as detailed below:

1) Model extensionsa) Are there examples of enhancers in the gap gene network, in which caudal acts as both an activator and a repressor for a given target gene? Providing evidence for such regulation would make the work less speculative.

b) Information on how the hs:hb experiment was modeled is missing. Importantly, ectopic hb does not contain enhancers that are responsive to the other genes and therefore should not be regulated by them. This appears to be an essential flaw in the model.

c) The Materials and methods section doesn't contain details about how the gradient G was modeled. Since this is integral to the understanding of the paper, please add this important information.

d) In Figure 6, why is mlpt localized posteriorly with uniform caudal (i.e. axn RNAi)? In my simulations of the model with a domain size of length 1E-4 and G=1.5 (rather than G=1.5*exp(-x/0.3E-4), which recapitulates the modeling results in this paper), the concentrations of all network constituents are uniform (with X_5 being the highest and X_4 or X_3, presumably representing mlpt, being very low) rather than posteriorly localized. Please clarify this discrepancy with additional simulations.

e) From a theoretical point of view, this paper is very similar in many ways to a previous PNAS paper from the El-Sherif lab (even containing the same typo in the dX_D2/dt equation appearing in the subsection “Speed Regulation GRN (Module Switching model)” of the current paper, i.e. X_3 in the fourth term of the second equation should be X_4). To add novelty to the model, the paper could benefit from an extension to analyze dynamic gradient formation rather than assuming a static caudal protein gradient for the initial phases of patterning.

2) Experimental extensions and methodological detailsa) If feasible, the validity of the model should be further tested by repeating the hs:hb experiment in a caudal null mutant to assess whether patterning after hb activation is still dependent on the initial morphogen caudal.

b) Even a heatshock in WT embryos has a large influence on hb expression, which complicates the analysis. Are there milder heatshock conditions that do not influence normal development in WT embryos?

c) The information for the piggyback vector, which should appear in Figure 4—figure supplement 1, is missing. Please add this information.

d) The formula for standard error quantification in the Materials and methods section is unclear.

e) The axn dsRNA concentration is missing. Please add this information.

3) Manuscript organizationa) The authors claim in several instances that this model might also underlie patterning in "possibly all insects". Unless tested in at least one more organism, this claim should be toned down throughout the manuscript.

b) Several references are missing (e.g. Berghammer et al., 1999). Please correct this throughout the manuscript.

---

## [Author Response]

Essential revisions:1) Include a basic description of the HS-hb phenotype (or range of phenotypes) – i.e. describe the effects on segment number, segment identity, duration of segmentation, and germband morphology. The data visualization in Figures 4—figure supplement 3, Figure 5—figure supplement 2 and Figure 5—figure supplement 4 should also be improved.

We added a description of the cuticular and morphological phenotypes of heat-shocked hs-hb embryos to Appendix 1, including effects on segment number, segment identity, and germband morphology. Further analysis of the segmentation phenotype will be described in detail in future work. In this paper, we restricted our analysis to the gap gene level.

Visualization in Figure 4—figure supplement 3, Figure 5—figure supplement 2 and Figure 5—figure supplement 4 is now improved.

2) The paper should be rewritten to minimize content overlap with the PNAS paper, i.e. streamline everything before "A dual response upon re-inducing hunchback in the Tribolium embryo", and appropriately discuss previous studies.

Now we shortened our theoretical sections. In addition, for brevity, we removed the analysis of the AC-DC circuit (and we removed the corresponding Supplementary Figure 1 in the original manuscript).

3) Address the axin RNAi results with additional simulations (i.e. show the simulation results for the axin RNAi experiment), and examine gap gene expression after cad RNAi (using late stage knockdown if appropriate).

In axn RNAi embryos, cad gradient extends to cover the whole embryo (but still expressed in a gradient, nonetheless). This is now mentioned in the section “The re-induction of gap gene sequence upon re-inducing hb is specific to the active-zone” of the revised manuscript. In addition, we added a quantification of the cad gradient in WT and axn RNAi embryos to Figure 6—figure supplement 1 of the revised manuscript. Simulations of axn phenotype (with and without pulsed hb re-activation) are as well added to the manuscript (Video 12) and described in the documentation of computational modeling (now moved to Appendix 1).

cad RNAi, unfortunately, results in very low fecundity; so it is not possible to do the same type of analysis done in our work for cad RNAi. Anyway, strong cad RNAi embryos are completely transformed into head fates (not patterned by the gap gene expression that we are analyzing here). This could be circumvented by late embryonic injection of dsRNA in order to skip the head/trunk fate determination phase. However, performing late RNAi perturbations by embryonic injection is challenging in Tribolium. Alternative techniques like applying dsRNA + electroporation to germband cultures is needed. However, such methods are still under development.

4) Add missing details regarding the methodology (e.g. dsRNA concentration, gradient modeling, morphological landmarks to define the three embryonic regions, etc..)

dsRNA concentration is now provided.

Details of gradient modeling is now added to the computational modeling documentation (computational modeling in Appendix 1). In addition, we provided Matlab codes for all of our simulations (Supplementary file 1).

We added the morphological landmarks we used to define the three embryonic regions: Active-Zone, Anterior, and HS-Anterior in the Materials and methods section under “Quantification of gene expressions in HS Anterior, Anterior, and Active-Zone”.

Strongly encouraged revisions:1) Repeat the hs:hb experiment in a caudal loss-of-function background (mutant or RNAi) to assess whether patterning after hb activation is still dependent on the initial morphogen caudal.

See our answer to point (3) of “Essential revisions” above.

2) Provide examples of enhancers in the gap gene network, in which caudal acts as both an activator and a repressor for a given target gene to make the work less speculative.

See our answer to point (3) of “Essential revisions” above.

Additional desirable revisions:1) Optimize the heatshock conditions to make the experiments interpretable in the framework of the speed-regulation model.

Unfortunately, it was not possible to generate a series of hs-hb perturbations of varying strengths that would give a homogenous response in populations of embryos.

2) To add novelty to the model, the paper could benefit from an extension to analyze dynamic gradient formation rather than assuming a static caudal protein gradient for the initial phases of patterning.

In the revised manuscript, we added simulations of dynamic gradients applied to the French flag and the Speed Regulation models (Videos 2 and 5 of the revised manuscript). In addition, we provided modeling methods (computational modeling, Appendix 1) and Matlab code for our simulations in which we added an option to adjust gradient buildup dynamics as needed (Supplementary file 1).

3) Tone down the relevance for patterning in other insects unless additional evidence is provided.

Done.

It would be interesting for the authors to comment on the relationships between their own findings and those of Hajime Nakao in the Bombyx embryo (Nakao 2016). That study implicated Hb in the termination of segment addition and Hox gene expression, rather than reinitiating it.

We now discuss this point in the Discussion of the revised manuscript.

In addition, please consider all points in the individual reviews below.

Reviewer #1:

[…] 1) The manuscript takes a long time (nearly 200 lines) to home in on the question actually addressed by the experiments – that of a genetic cascade vs a concentration dependent timer gene. I think that readability, focus, and clarity would all be improved by streamlining the paper and removing/drastically shortening the long introductory diversion into models of patterning mediated by static morphogen gradients.

We now shortened the Introduction. We would like to point out, though, that an important component of our manuscript (at least to us) is to highlight and contrast the difference between the French Flag model and the Speed Regulation model. Specifically, we wanted to show that the dynamics of the French Flag model could be very similar to the Speed Regulation model, and hence, we need other measures to differentiate between the two models, as exemplified by the heat-shock perturbation we are documenting in the manuscript. This point is not discussed in Francois and Siggia 2010, and we needed to describe the essence of that work to make the comparison clear.

Aside from being beside the point, much of this material simply repeats the authors' 2017 PNAS paper, or re-describes points already made in Francois and Siggia 2010. (Indeed, Figures 1B, 1D, 2A, 2B, and 3A all reproduce images from the 2017 paper, without remark.)

We now pointed out that these are adapted from Zhu et al., 2017. We find it very difficult to discuss the ideas presented in the current manuscript without referring to some of the models in Zhu et al., 2017.

In addition, I am concerned about a misleading presentation of Drosophila gap gene regulation as being of the "French Flag" type, instead of discussing the authors' models in the context of published work on emergent gap gene dynamics from the Reinitz, Jaeger, and Gregor labs.

This part is omitted in the revised manuscript.

2) I find some of the terminology used when discussing the models to be unhelpful – particular the comparison between "French flag"/"threshold-based" scenarios and the authors' "speed-regulation" model. First, the key hypothesis being tested is the presence of a concentration dependent timing factor in the "active" zone, vs timing information emerging from the interplay of the gap factors themselves. (I have used the terms "timer gene" and "genetic cascade" in this review to refer to these respective concepts.) While the authors repeatedly state that their findings validate their "speed regulation" model (e.g. subsection “A dual response upon re-inducing hunchback in the Tribolium embryo”, fifth paragraph, Discussion, third and last paragraphs), this claim is potentially misleading as they instead simply rule out the "timer gene" hypothesis; the experimental results are not informative regarding the "speed regulation" aspect of the speed regulation model.

We modified our wording in the revised manuscript, stating that our results are ‘consistent with’ the speed regulation model instead of ‘validating it’, since our simulations based on the speed regulation model recapitulate the heat-shock experiments.

Second, the use of "threshold-free" to refer to the absence of a concentration-dependent timer gene is potentially confusing, because this "threshold-free" scenario doesn't rule out the existence of other regulatory thresholds within the genetic network.

By ‘threshold-free’ we mean thresholds on the morphogen gradients, not other regulatory interactions within the network. We clarified this in the revised manuscript.

Third, the authors use the French flag model as a reference point when introducing their models. However, they also argue for a redefinition of the French flag (Introduction, second paragraph) that fits with their GRN implementations. Given that their study system isn't well represented by a French flag scenario anyway, this strategy seems unnecessarily complicated and potentially confusing.3) While the paper is understandably focused on the effects on gap gene expression, it should also include some basic description of the terminal phenotypes resulting from the HS-Hb treatments. There is a line in the Materials and methods section (subsection “Generation of transgenic beetles”) about larvae developing homeotic transformations and/or additional trunk segments. This strikes me as very interesting (and increases support for the authors' model) and I would have ideally liked to have seen some characterisation of these effects: for example, cuticle preparations and in situs against Hox genes and segmentation genes. In addition, the HS-Hb germbands shown in Figure 4—figure supplement 2, Figure 5—figure supplement 1, and Figure 5—figure supplement 1 are clearly much shorter than their HS-wt counterparts (whereas the model suggests that they ought to generate more segments). This is also interesting but not remarked upon. Is the morphological difference due to effects on convergent extension, cell proliferation, or both? And, if these short germbands eventually produce larvae with extra segments, is the duration of segmentation/axial extension therefore significantly protracted?

See our response to point (1) of “Essential Revisions” in the Editor’s summary above.

Reviewer #2:

In this manuscript, Boos et al. propose that the anterior-posterior axis in Tribolium is regulated by a threshold-free speed regulation mechanism based on a gradient of caudal and gap gene interactions. This is largely based on the observation that pulsed hb expression re-initiates the patterning cascade, which is roughly consistent with the predictions from a model of the gap gene network. The experimental observation is very interesting, but the model in its current state remains speculative. The study therefore needs to be extended to further test the validity of the model as detailed below:1) Model extensionsa) Are there examples of enhancers in the gap gene network, in which caudal acts as both an activator and a repressor for a given target gene? Providing evidence for such regulation would make the work less speculative.

See our response to point (2) of “Strongly encouraged revisions” in the Editor’s summary above.

b) Information on how the hs:hb experiment was modeled is missing. Importantly, ectopic hb does not contain enhancers that are responsive to the other genes and therefore should not be regulated by them. This appears to be an essential flaw in the model.

See our response to point (2) of “Strongly encouraged revisions” in the Editor’s summary above.

c) The Materials and methods section doesn't contain details about how the gradient G was modeled. Since this is integral to the understanding of the paper, please add this important information.

Details of gradient modeling is now added to the computational modeling documentation (Appendix 1). In addition, we provided Matlab codes for all of our simulations (Supplementary file 1).

d) In Figure 6, why is mlpt localized posteriorly with uniform caudal (i.e. axn RNAi)? In my simulations of the model with a domain size of length 1E-4 and G=1.5 (rather than G=1.5*exp(-x/0.3E-4), which recapitulates the modeling results in this paper), the concentrations of all network constituents are uniform (with X_5 being the highest and X_4 or X_3, presumably representing mlpt, being very low) rather than posteriorly localized. Please clarify this discrepancy with additional simulations.

See our response to point (3) of “Essential Revisions” in the Editor’s summary above.

The typo is fixed.

e) From a theoretical point of view, this paper is very similar in many ways to a previous PNAS paper from the El-Sherif lab (even containing the same typo in the dX_D2/dt equation appearing in the subsection “Speed Regulation GRN (Module Switching model)” of the current paper, i.e. X_3 in the fourth term of the second equation should be X_4). To add novelty to the model, the paper could benefit from an extension to analyze dynamic gradient formation rather than assuming a static caudal protein gradient for the initial phases of patterning.

See our response to point (2) of “Additional desirable revisions” in the Editor’s summary above.

2) Experimental extensions and methodological detailsa) If feasible, the validity of the model should be further tested by repeating the hs:hb experiment in a caudal null mutant to assess whether patterning after hb activation is still dependent on the initial morphogen caudal.

See our response to point (3) of “Essential revisions” in the Editor’s summary above.

b) Even a heatshock in WT embryos has a large influence on hb expression, which complicates the analysis. Are there milder heatshock conditions that do not influence normal development in WT embryos?

See our response to point (1) of “Additional desirable revisions” in the Editor’s summary above.

c) The information for the piggyback vector, which should appear in Figure 4—figure supplement 1, is missing. Please add this information.

Now added (Figure 4—figure supplement 1).

d) The formula for standard error quantification in the Materials and methods section is unclear.

Done.

e) The axn dsRNA concentration is missing. Please add this information.

Now added.

3) Manuscript organizationa) The authors claim in several instances that this model might also underlie patterning in "possibly all insects". Unless tested in at least one more organism, this claim should be toned down throughout the manuscript.

Done.

b) Several references are missing (e.g. Berghammer et al., 1999). Please correct this throughout the manuscript.

Done.